# Magnesium hexacyanoferrate nanocatalysts attenuate chemodrug-induced cardiotoxicity through an anti-apoptosis mechanism driven by modulation of ferrous iron

Minfeng Huo [1,2,3,8], Zhimin Tang[4,5,8], Liying Wang[1], Linlin Zhang[2], Haiyan Guo[6], Yu Chen [7], Ping Gu [3,4] ✉ & Jianlin Shi [1,2,3] ✉

Distressing and lethal cardiotoxicity is one of the major severe side effects of using anthracycline drugs such as doxorubicin for cancer chemotherapy. The currently available strategy to counteract these side effects relies on the administration of cardioprotective agents such as Dexrazoxane, which unfortunately has unsatisfactory efficacy and produces secondary myelosuppression. In the present work, aiming to target the characteristic ferrous iron overload in the doxorubicin-contaminated cardiac microenvironment, a biocompatible nanomedicine prepared by the polyvinylpyrrolidone-directed assembly of magnesium hexacyanoferrate nanocatalysts is designed and constructed for highly efficient intracellular ferrous ion capture and antioxidation. The synthesized magnesium hexacyanoferrate nanocatalysts display prominent superoxide radical dismutation and catalytic $H_2O_2$ decomposition activities to eliminate cytotoxic radical species. Excellent in vitro and in vivo cardioprotection from these magnesium hexacyanoferrate nanocatalysts are demonstrated, and the underlying intracellular ferrous ion traffic regulation mechanism has been explored in detail. The marked cardioprotective effect and biocompatibility render these magnesium hexacyanoferrate nanocatalysts to be highly promising and clinically transformable cardioprotective agents that can be employed during cancer treatment.

Catalytic antioxidation is one of the most attractive biomedical frontiers and involves the reversal of comprehensive pathological processes, including cancer development, systemic toxicity, inflammation, and metabolic degeneration[1–3]. One of the major pathological origins has been identified as the abnormal burst/accumulation of oxidative stress caused by reactive oxygen species (ROS) and/or metal ions. Such oxidative stresses and the associated transcriptional and proteomic alterations collectively lead to damage and the destruction of certain functioning cells and tissues and normal

physiological processes in living organisms, inducing a variety of diseases and pathologies[4–6]. Cardiotoxicity is one of the most distressing and life-threatening systemic pathologies that is frequently induced as a harmful side effect in cancer patients receiving anthracycline chemotherapeutics (e.g., doxorubicin (DOX), idarubicin), antimetabolic antineoplastic agents (e.g., 5-fluorouracil), or antimicrotubular drugs (e.g., paclitaxel)[7]. Specifically, significant dose-dependent DOX-induced cardiotoxicity (DIC) and eventually, irreversible congestive heart failure, have been observed in the clinic, and the therapeutic

performance of these agents and patient prognosis is hence largely counteracted or deteriorated[8,9]. Such distressing side effects have made the traditional and commonly used chemodrug DOX carry the bad reputation of being a red devil in the medical community[10].

The fundamental biological consequences of DIC have long been regarded to be due to the accumulation of oxidative stress inside cardiomyocytes, where DOX molecules are supposed to bind to intracellular DNA under activation by flavin reductase, which reduces DOX molecules to semiquinone compounds that can further react with oxygen to form superoxide anion radicals, $H_2O_2$ and hydroxyl radical species[11]. However, the clinical failure of the application of *N*-acetylcysteine (a direct ROS trapping agent) for cardioprotection has denied the oxidative stress-induced hypothesis. Recent pioneering studies have suggested that DOX-contaminated heart tissue exhibits a prototypical abnormal accumulation of intracellular ferrous species that is accompanied by the associated generation of oxidative stress[12]. This discovery of the cardiotoxic pathological origin of ferrous iron overload, in addition to secondary oxidative stress, offers a significant therapeutic target for capturing accumulated intracellular ferrous species to prevent DIC[13]. Thus, it was conceived that better cardioprotection and prognosis could be achieved by simultaneously removing pathological iron species and reducing the accompanying oxidative stress levels.

Dexrazoxane (DXZ), as the only iron capture agent that is clinically used for cardioprotection, can effectively chelate intracellular ferrous ions with its ketone groups to prevent the ferrous-dependent Fenton reaction. However, DXZ suffers from a limited number of ferrous species chelation sites. In addition, DXZ has also been questioned for its secondary side effect of myelosuppression and interference with antitumour therapeutics[11]. Recently, Kim and co-workers developed ε-poly-L-lysine-directed multi-deferoxamine-conjugated nanochelators with greatly improved pathological ferrous ion clearance performance[14]. Nevertheless, their chelation effects for ferrous ions are not strong enough compared to chemical capture into the hexacyanoferrate lattice.

Nanocatalysts with multienzyme-like antioxidative activity have been designed and employed in living organisms to treat a variety of pathologies in biomedicine, such as reperfusion-induced injury, acute kidney injury and inflammatory bowel diseases[15–17]. Owing to their specific electronic structure, these nanocatalysts, such as $Co^{2+}$-containing metal-organic frameworks[18], iron-containing carbon nanozyme[19] and the $Cu^{2+}$-containing tannic acid nanosheets[20], show excellent and stable antioxidative enzymatic catalytic activities mimicking superoxide dismutase (SOD) and catalase (CAT). Ferric hexacyanoferrate ($Fe_4[Fe(CN)_6]_3$), also known as Prussian blue (PB), has been approved by the U.S. Food and Drug Administration as a detoxification agent to treat radioactive caesium (Cs) or thallium (Tl) contamination in patients with guaranteed in vivo biocompatibility and biosafety. PB nanoparticles have been engineered into various structures to serve as nanotherapeutic agents for drug delivery[21,22], antioxidation, etc.[23,24]. Nanoparticles of the PB analogue PBA have also been synthesized as multifunctional nanotherapeutics by substituting non-iron species for ferrous and/or ferrite species[25,26]. Interestingly, PBA nanoparticles are highly potent iron chelators, as iron-based PB features a low solubility product constant of $3.3 \times 10^{-41}$, which is far lower than those of its non-iron-based counterparts, endowing the PBA nanoparticles with the prominent capability of capturing ferrous species by reforming the initial PB nanoparticles with exceptionally high kinetics[26] displaying great potential and broad prospects in biomedical applications.

In the present work, we report a polyvinylpyrrolidone-assembled magnesium hexacyanoferrate nanocatalytic medicine (MgHCF NCs) for the effective intracardial elimination of ferrous species and antioxidative cardioprotection against DIC (Fig. 1). We show that synthetic MgHCF NCs can effectively bind pathological ferrous species and attenuate oxidative stresses both in vitro and in vivo, which is accompanied by specific transcriptional and proteomic alterations associated with abnormal ferrous ion traffic. The biocompatibility, biosafety and in vivo cardioprotection performance are investigated. The present paradigm demonstrates substantial body toxicity alleviation and cardioprotection by MgHCF NCs administration during cancer treatment with the chemodrug DOX.

## Results
### Synthesis and characterization of MgHCF NCs
MgHCF NCs were synthesized via a facile self-assembly method under the direction of polyvinylpyrrolidone (PVP, MW = 10 kDa) (Fig. 2a). The addition of potassium ferricyanide ($K_3[Fe(CN)_6]$) into the solution containing an equimolar amount of $Mg^{2+}$ simultaneously led to the formation of a pale-yellow aqueous solution. The products were then purified by dialysis and desiccated in vacuo. This process yielded MgHCF NCs with a dark-green appearance (Supplementary Fig. 1). The synthetic MgHCF NCs were used upon rehydration with saline or PBS at the desired concentrations and doses. The obtained nanocatalysts

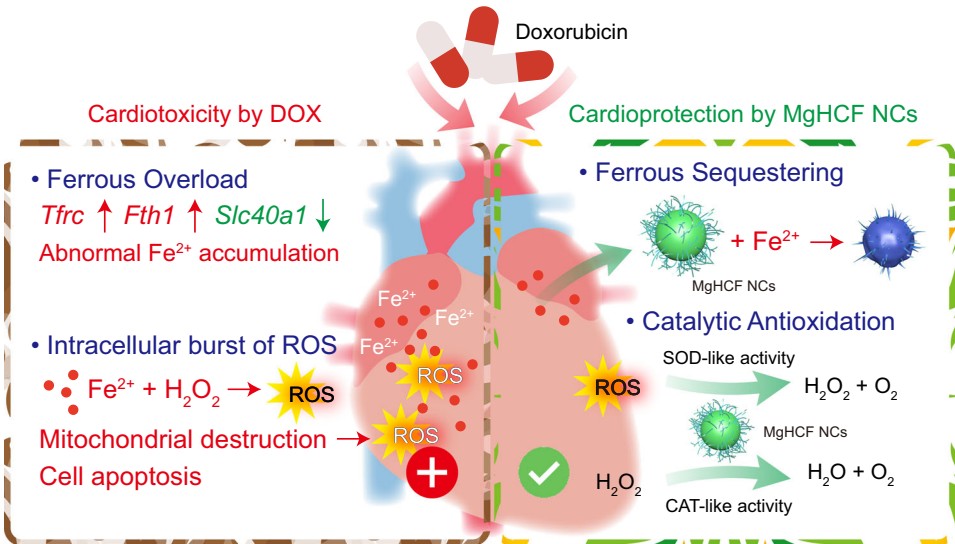

**Fig. 1 | Scheme of cardioprotective strategy by MgHCF NCs.** Schematic cardiotoxicity generation during chemotherapy using doxorubicin as well as herein proposed cardioprotective strategy by the synthetic MgHCF NCs.

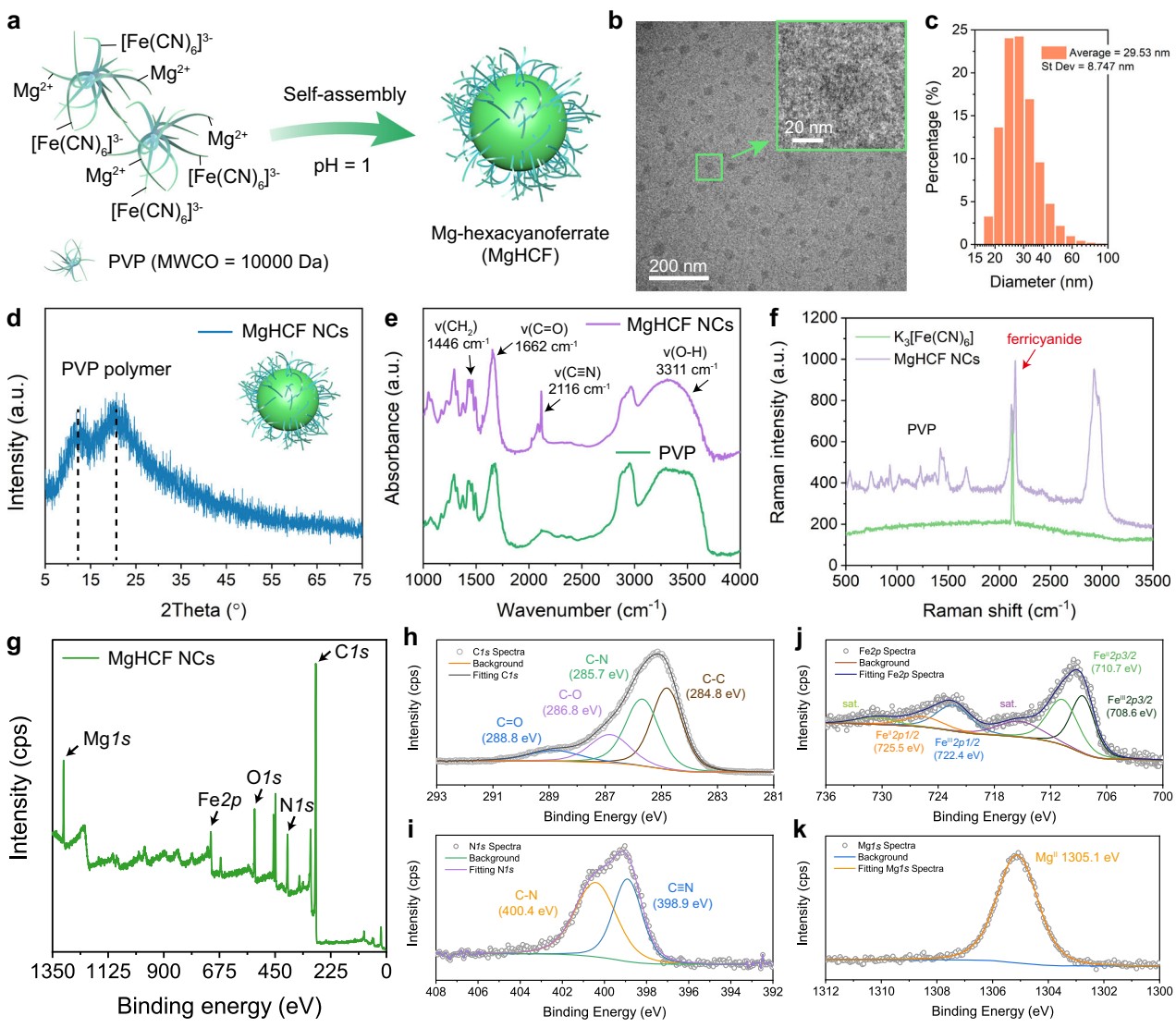

**Fig. 2 | Synthesis and material characterizations of MgHCF NCs. a** Schematic illustration of the PVP-directed self-assembly of MgHCF NCs. **b** TEM image of the MgHCF NCs. The inset shows a typical MgHCF nanoparticle. **c** Hydrodynamic particle diameter distribution of MgHCF NCs suspended in an aqueous solution. **d–g** XRD pattern (**d**), FTIR spectra (**e**), Raman spectra (**f**) and XPS spectra (**g**) of MgHCF NCs, PVP or $K_3[Fe(CN)_6]$. **h–k** High-resolution elemental XPS spectra for C (**h**), N (**i**), Fe (**j**) and Mg (**k**).

showed a uniform particulate morphology with an average particle diameter of $28.98 \pm 7.73$ nm, which was calculated by measuring one hundred NPs by transmission electron microscopy (Fig. 2b, Supplementary Fig. 2). According to the dynamic light scattering method, the hydrodynamic diameters of the MgHCF NCs were determined to be $29.53 \pm 8.75$ and $31.51 \pm 8.67$ nm in aqueous solution and saline, with polydispersity indices (PDIs) of 0.39 and 0.27, respectively (Fig. 2c, Supplementary Fig. 3a). The zeta potential of the MgHCF NCs aqueous suspension was determined to be $-19.3 \pm 3.91$ mV and showed excellent physiological stability for up to 2 months according to the time-course zeta potential profile (Supplementary Fig. 3b). From the X-ray diffraction pattern of the MgHCF powder, characteristic peaks of the PVP polymer were observed (Fig. 2d, Supplementary Fig. 4a). We also confirmed the crystalline nature of the precursor $K_3[Fe(CN)_6]$ as well as PB NCs synthesized in the presence of MgHCF NCs and free ferrous ions (Supplementary Fig. 4b). We then acquired FT-IR spectra for the synthetic MgHCF nanocatalysts and PVP polymer and found that the $-CH_2$ wagging motion had shifted to the relatively higher frequency of 1446 cm⁻¹, suggesting the presence of an interaction between the positively charged C–H bonds and negatively charged ferricyanide

molecules (Fig. 2e)[27]. Characteristic Raman shift signals from 500 to 1800 cm⁻¹ from the PVP polymer were identified in the Raman spectrum of the synthetic NCs[28]. Additionally, C≡N stretching vibrations were apparent, revealing the presence of the metallic cyanide structure (Fig. 2f)[29]. Due to the interactions between PVP–Mg²⁺ as well as PVP–ferricyanide ions, we, therefore, confirmed that the architecture of the MgHCF NCs is amorphous and assembled by ferricyanide ions and magnesium ions interacting with the polyvinyl pyrrolidone polymer. Powder samples of MgHCF NCs were also subjected to X-ray photoelectron spectroscopy analyses, which indicated the coexistence of C, N, O, Fe and Mg elements in the NCs at atomic percentages of 75.18%, 9.84%, 10.39%, 2.02% and 2.57%, respectively (Fig. 2g). From the high-resolution scanning spectra, C1s could be deconvoluted into C–C bonding (284.8 eV), C–N bonding (285.7 eV), C–O bonding (286.8 eV) and C=O bonding (288.8 eV), revealing the presence of polyvinylpyrrolidone (Fig. 2h). The N1s spectrum could be deconvoluted into C–N bonding (400.4 eV) and C≡N bonding (398.9 eV), which was attributable to the cyano groups of the MgHCF NCs (Fig. 2i). According to the Fe2p and Mg1s spectra, the presence of multivalent iron (calibrated concentration ratio of $Fe^{III}:Fe^{II} = 1:1.25$) and $Mg^{II}$ species was

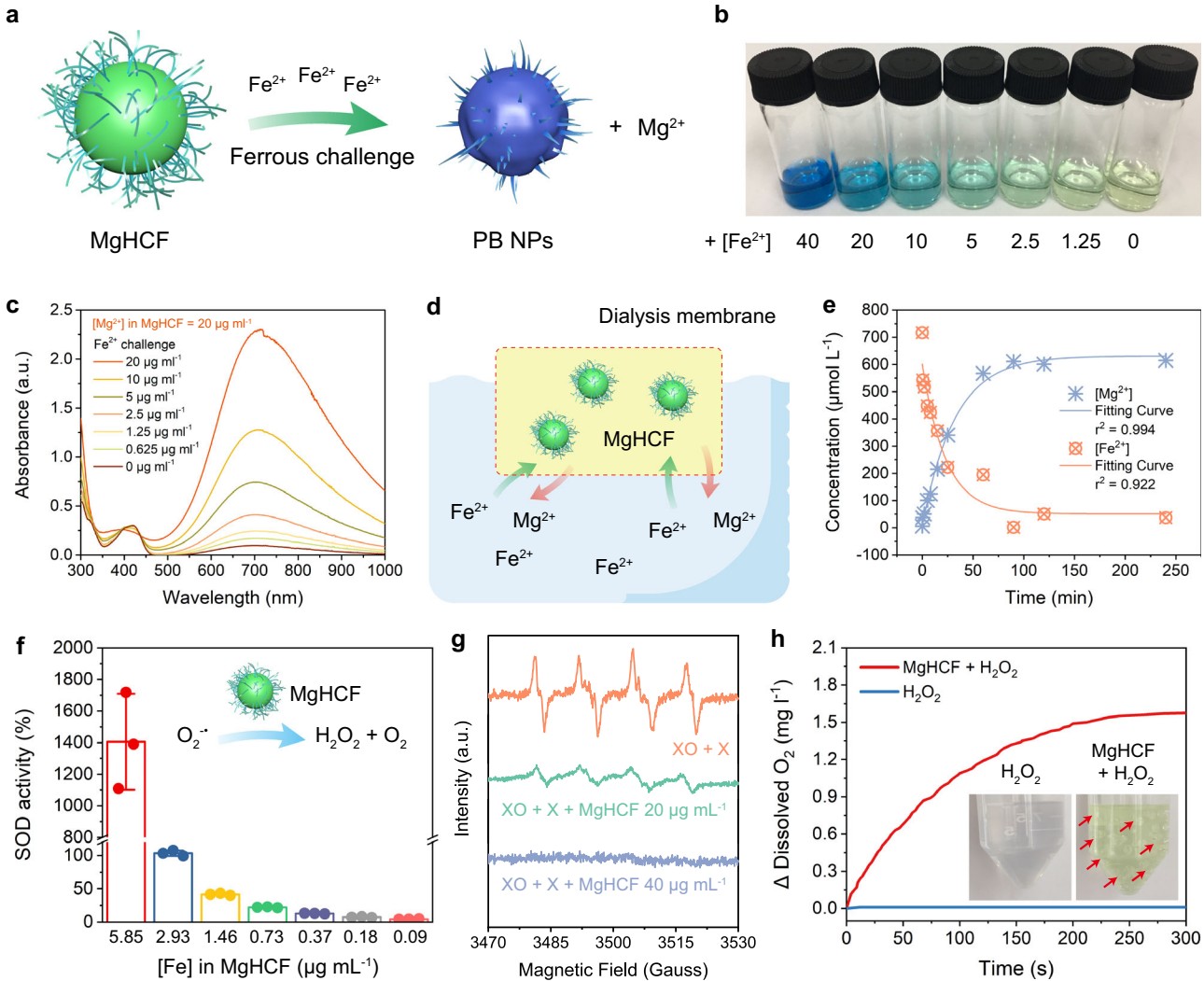

**Fig. 3 | Ferrous capturing and antioxidation performance of MgHCF NCs.**
**a** Schematic illustration of the ferrous binding activity of MgHCF as a Prussian blue analogue by recovering to the pristine Prussian blue nanoparticles. **b** Digital photograph of the aqueous solutions containing MgHCF NCs with the additions of the ferrous ions at varying concentrations. **c** Corresponding UV–vis spectra of respective samples in (**b**). **d** Schematic illustration of the dialysis-membrane separation-regulated cation exchange for MgHCF NCs. **e** Time-dependent elemental concentration profiles for $Mg^{2+}$ and $Fe^{2+}$ in the solution during the dialysis-membrane separation-regulated cation exchange for MgHCF NCs. **f** Superoxide radical dismutation activities of MgHCF at varied [Fe]. $n = 3$, Data are presented as mean ± s.d. **g** ESR spectra of MgHCF in eliminating the superoxide anions generated in xanthine oxidase + xanthine system in the presence of BMPO as the radical trapper. **h** Variation of the dissolved oxygen levels in a $H_2O_2$ (2 mM) solution in the presence or absence of MgHCF. The inset shows the digital photograph of samples containing $H_2O_2$ or MgHCF + $H_2O_2$. Oxygen bubbles are indicated by red arrows.

confirmed (Fig. 2j–k). From the above material characterizations, we believe that the MgHCF NCs are amorphous nanoparticles without crystallized Prussian blue or analogue structures. The NCs assembled among $Mg^{2+}$, hexacyanoferrate and PVP polymers through electrostatic interactions and are physiologically stable for ferrous ion binding and antioxidation both in vitro and in vivo.

### Ferrous ion capture and the multicatalytic activities of MgHCF NCs

The synthetic MgHCF NCs feature a strong affinity for ferrous ions to form the original PB nanostructure while simultaneously releasing $Mg^{2+}$ from the structure of MgHCF via displacement by ferrous species in the nanostructure (Fig. 3a). The cation exchange performance of the MgHCF NCs towards ferrous ions was investigated in free and dialysis membrane-confined conditions. The addition of varying concentrations of $Fe^{2+}$ (0–40 μg ml$^{-1}$) in the form of iron(II) ammonium sulfate to the pale yellow MgHCF NCs suspension (40 μg ml$^{-1}$) immediately led to a change in the optical absorption and colour from yellow to green and

finally blue (Fig. 3b). Equilibrium was reached in 30 s. The mixture of MgHCF NCs upon challenge with varying doses of the ferrous species was further assayed by UV–vis spectroscopy. The primitive MgHCF NCs aqueous suspension displayed its characteristic optical absorption positioned ~425 nm. Once challenged with ferrous species of increasing concentrations, the absorption peak of the MgHCF NCs aqueous suspension at 425 nm gradually weakened, while a broad optical absorption peak from 500 to 1000 nm (peaking at 700 nm) emerged and intensified, suggesting the intervalence charge transfer between $Fe^{2+}$ and $Fe^{3+}$ [30], directly revealing the formation of the PB structure (Fig. 3c). To monitor the ionic concentration during the cationic exchange between MgHCF NCs and $Fe^{2+}$, dialysis membrane sealed MgHCF NCs (10 mL, [$Mg^{2+}$] = 20 μg mL$^{-1}$) were placed into a solution containing excess ferrous species. At predetermined time points, the concentrations of iron and magnesium were determined by inductively coupled plasma–optical emission spectrometry (ICP–OES) (Fig. 3d). Upon cationic exchange, the iron concentration rapidly decreased within 65 min, while the magnesium concentration

gradually increased in nearly identical molar concentrations (Fig. 3e). These experiments specifically indicated that the $Mg^{2+}$ in the MgHCF NCs can be well-displaced by the ferrous ions that have a much higher binding affinity towards the PB architecture, which enables the chemical binding of the ferrous species by the MgHCF NCs and simultaneous release of magnesium ions with fast kinetics. Even with interference from intracellular biomacromolecules (full RPMI medium with fetal bovine serum), the cationic exchange performance was unaffected (Supplementary Fig. 5).

We next investigated the antioxidative multicatalytic performance of MgHCF NCs, focusing on radical elimination reactions such as superoxide anion radicals and $H_2O_2$. Using a superoxide dismutase (SOD) activity kit, we found that the synthetic MgHCF NCs exhibited prominent catalytic activity to suppress superoxide-propagated reactions. Specifically, NCs at an iron concentration of $5.85\,\mu g\,mL^{-1}$ generated a relative radical-clearing activity of $1405 \pm 304\%$, while the half concentration of NCs exhibited a relative radical-clearing activity of $103.35 \pm 4.18\%$, suggesting powerful superoxide radical dismutation activity against superoxide anions (Fig. 3f). Then, the electron spin resonance (ESR) technique was employed to investigate the superoxide anion (stimulated by the xanthine oxidase + xanthine system) clearance efficiency by MgHCF NCs. Using BMPO as the radical trapping agent, the xanthine oxidase + xanthine system produced prominent spectral signals of superoxide anions, while coincubation with MgHCF NCs ($[Fe] = 20\,\mu g\,mL^{-1}$) effectively led to a reduction in the amplitude of the radical adducts, and a higher dose of the NCs ($[Fe] = 40\,\mu g\,mL^{-1}$) eliminated the superoxide anions completely (Fig. 3g). The prominent superoxide radical dismutation activity of the MgHCF NCs possibly originates from the low reduction potential of HCF during antioxidation[24]. Considering that superoxide anions are usually the primary radicals generated in living systems by systemic oxidative stresses and damage[31], MgHCF NCs are expected to offer antioxidative activity both in vitro and in vivo. In addition to superoxide anions, MgHCF NCs displayed catalytic $H_2O_2$ decomposition activity by decomposing $H_2O_2$ to form oxygen, a key process for complete antioxidation. We employed a dissolved oxygen electrode to measure an aqueous suspension containing MgHCF NCs in the presence or absence of $H_2O_2$ (2 mM). Significant and steady growth of the amount of dissolved oxygen was observed compared to the control in 300 s (Fig. 3h). Emerging $O_2$ bubbles in the tubes of the assay sample also support the catalytic $H_2O_2$ decomposition performance of MgHCF NCs.

Upon ferrous ion challenge with MgHCF NCs, PB NPs were immediately formed. It is therefore interesting to further evaluate the antioxidative performance of the postformed PB NPs after the ferrous ion binding of MgHCF NCs. To our surprise, the postformed PB NPs exhibited steady $H_2O_2$ decomposition performance at doses of 1.46, 2.93 and $5.85\,\mu g\,ml^{-1}$ PB NPs, generating a maximal dissolved $O_2$ concentration of $0.8\,mg\,l^{-1}$ in 300 s upon coincubation with $H_2O_2$ (Supplementary Fig. 6a). In addition, the superoxide radical dismutation activity of the PB NPs at the dose of $2.93\,\mu g\,ml^{-1}$ was determined to be 48.95%, accounting for 47% of the performance of the primitive MgHCF NCs at the identical concentration (Supplementary Fig. 6b). These data show that after the binding of ferrous ions to the MgHCF NCs, the postformed PB NPs preserve the ability to decompose $H_2O_2$ and superoxide radicals for antioxidation.

## MgHCF NCs alleviate cardiotoxicity and the corresponding genetic distributions

The anticancer chemotherapeutic DOX has been demonstrated to impair cardiomyocytes, leading to lethal cardiotoxicity in the human body[7]. Herein, rat cardiomyoblasts (H9c2 cell line) were employed for in vitro cellular evaluations[32,33], and the cellular protective effects of the synthetic MgHCF NCs and a cardioprotective agent, dexrazoxane (DXZ), were investigated (Fig. 4a). The cytotoxicity of DOX against the

cardiomyoblasts was initially evaluated by fluorescence dual-staining (calcein-AM/PI). It was observed that DOX treatment at doses of $2.5–80\,\mu g\,ml^{-1}$ effectively killed a substantial proportion of the cardiomyoblasts, as reflected by the cells stained magenta (pseudocolour) by the PI dye (Fig. 4b), a result that was confirmed by CCK-8 assay. Dose-dependent viability of the cardiomyoblasts was observed, with the $EC_{50}$ value of DOX calculated to be $27.95 \pm 5.41\,\mu g\,ml^{-1}$ (Fig. 4c, Supplementary Fig. 7a). Cytocompatibility assays in H9c2 cells revealed that the MgHCF NCs are noncytotoxic at the doses applied, indicating the high biocompatibility of both the MgHCF NCs and DXZ (Supplementary Fig. 7b, c). We next evaluated the viability of DOX-contaminated cells ($20\,\mu g\,ml^{-1}$) co-incubated with different concentrations of MgHCF NCs and DXZ. It was found that the addition of MgHCF NCs, as well as DXZ, could rescue a major proportion of DOX-treated cardiomyoblasts ($RCV_{MgHCF\,NCs}\% = 94.47\%$, $RVC_{DXZ}\% = 72.35\%$). Especially in the case of MgHCF NCs coincubation, DOX cardiotoxicity could be effectively reversed at doses as low as $40\,\mu g\,ml^{-1}$, presenting the promising cardioprotective potential of the MgHCF NCs compared to the clinically applied DXZ (Fig. 4d). In addition to the rat cardiomyoblast cell line H9c2, we also employed human cardiomyocyte AC16 cells for in vitro cellular investigations (Supplementary Fig. 8a). Treatment with DOX at a dose of $10\,\mu g\,ml^{-1}$ induced observable toxicity to AC16 cardiomyocytes. Most of the cardiomyocytes died under the DOX challenge at increased doses of 20, 40 and $80\,\mu g\,ml^{-1}$ (Supplementary Fig. 8b). We then selected a dose of $20\,\mu g\,ml^{-1}$ for cytoprotection evaluation with varying concentrations of DXZ and MgHCF NCs (Supplementary Fig. 8c). The results in the dual-stained microscopic images suggested that the minimum doses of DXZ and MgHCF NCs needed to completely rescue the contaminated cardiomyocytes were 25 and $20\,\mu g\,ml^{-1}$, respectively, which is primarily consistent with the cellular investigation in H9c2 cardiomyoblasts (Supplementary Fig. 8d, e).

To explore the cardiotoxicity and cardioprotective effects at the transcriptional level, we carried out whole-transcriptome analysis by mRNA sequencing (mRNA-seq) cardiomyoblasts subjected to different treatments, including untreated (Unt), DOX treatment (DOX), combinational DOX + MgHCF treatment (D_MgHCF) and combinational DOX + DXZ treatment (D_DXZ). Throughout the transcriptome, 15,380 differentially expressed genes (DEGs) were qualified by unsupervised dimensionality reduction by principal component analysis (PCA), comprising PC1 and PC2, which collectively accounted for 83.48% of the differences (Fig. 4e). We found that PCA was able to distinguish between the Unt and DOX groups but not the DOX and D_DXZ groups, illustrating the similarity of the transcriptional alterations of the cells treated with DOX and D_DXZ. PCA also distinguished the D_MgHCF group from the other groups. The D_MgHCF samples showed higher transcriptional similarity to the Unt group than to the other groups, preliminarily indicating the cardioprotective effect of the MgHCF NCs at the mRNA transcriptional level (Fig. 4e). Based on the analyses from the hierarchical clustering method, all of the DEGs were classified into three subclusters to reveal the gene distribution trends compared to the untreated group (Fig. 4f). From the average genetic distribution of 4775 genes in subcluster I, the DOX and D_DXZ groups produced higher $\log_2(FPKM + 1)$ values than the Unt and D_MgHCF groups, suggesting that pathological transcriptional alterations were caused by DOX. Comparatively, the transcriptional alterations in the D_DXZ group were distributed similarly to the DOX group, while the D_MgHCF group specifically showed similarity to the Unt group, suggesting a better cardioprotective effect of MgHCF than the clinically applied DXZ. A total of 1159 genes were allocated in genetic subcluster II, in which DOX treatment led to a substantial pathological downregulation of the genes, while strong reversing effects against DOX were observed by both D_DXZ and D_MgHCF treatment. The remaining 9447 genes were assigned to the third genetic

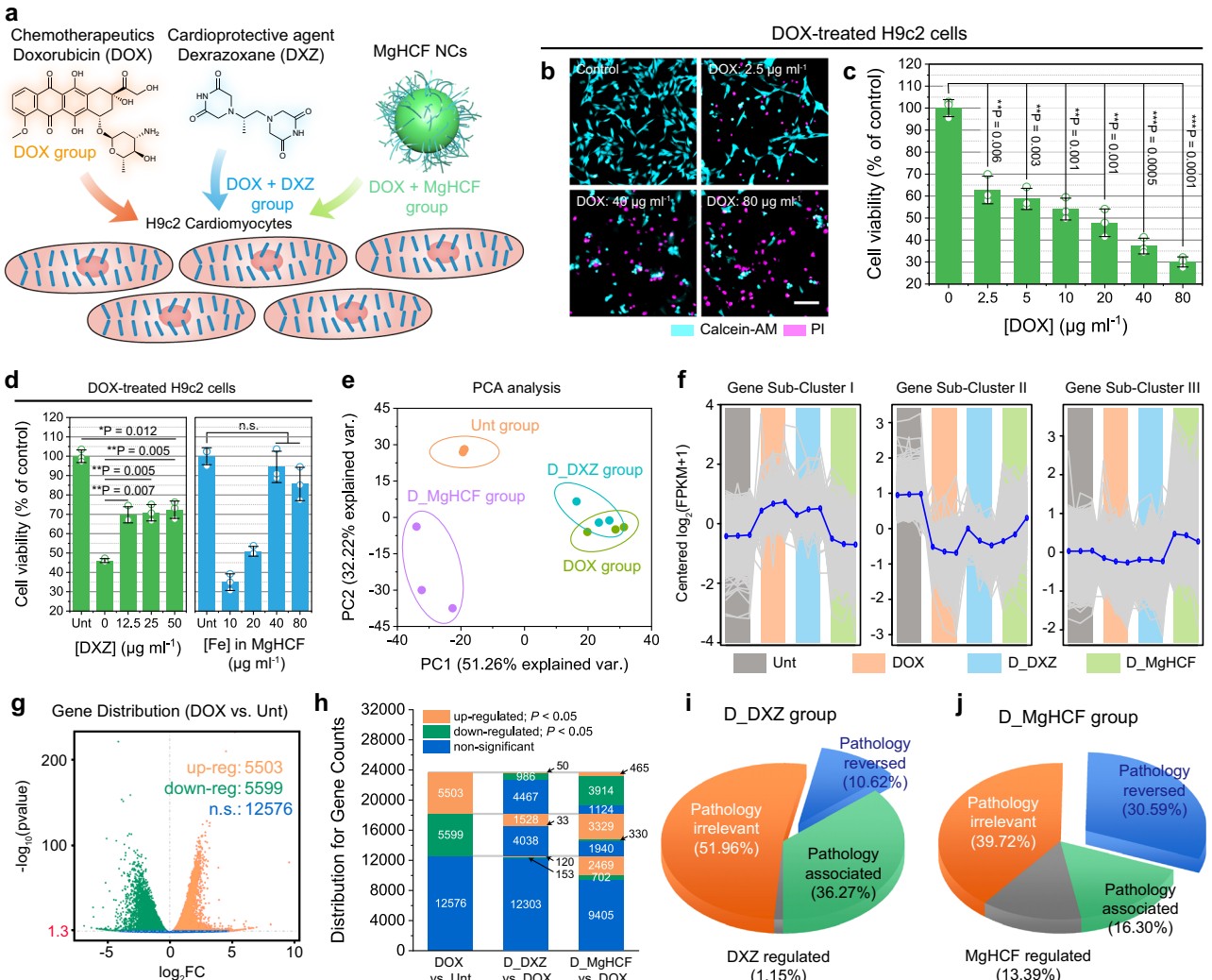

**Fig. 4 | In vitro cellular evaluations and high throughput mRNA sequencing.** **a** Schematic illustration of the in vitro cellular experiments. **b** Confocal microscopic images of Calcein-AM/PI-stained cells treated with varying doses of DOX. Scale bar: 50 μm. **c** Viability of H9c2 cells treated with varying doses of DOX ($n = 3$ of biological replicates). Data are presented as mean ± s.d. Statistical significance is assessed by Student $t$'s one-tailed test. **$P < 0.01$, ***$P < 0.001$. **d** Viability of the H9c2 cells treated with DOX supplemented with different concentrations of DXZ or MgHCF ($n = 3$ of biological replicates). Data are presented as mean ± s.d. Statistical significance is assessed by Student $t$'s one-tailed test. *$P < 0.05$, **$P < 0.01$. n.s. for non-significant. **e** PCA analysis for the cell samples in different groups. Data are presented as they are. **f** Sub-clustering indications (by H-clustering method) for the mRNA whole genetic regulation distributions within cells in different groups. Data are presented as they are with the mean value shown in blue. **g** Volcano plots for the DEG distributions between DOX and Unt groups in terms of up-regulated, down-regulated, and non-significant genes. Significant tests are based on the negative binomial distribution. *$P < 0.05$ is equivalent to the $-\log_{10}$(pvalue) >1.3. Data are presented as they are. **h** Distributions of gene counts in DOX vs. Unt for pairwise comparisons between D_MgHCF and DOX, and between D_DXZ and DOX. **i** and **j** Gene percentages for D_DXZ group (**i**) and D_MgHCF group (**j**) following gene grouping.

subcluster, in which cells treated with DOX + MgHCF displayed the upregulation of genes compared to the Unt, DOX and D_DXZ groups (Fig. 4f).

The genetic distributions in the different treatment groups are shown in the volcano plots for pairwise comparison, with the upregulated, downregulated, and nonsignificant genes indicated (Fig. 4g, Supplementary Figs. 9, 10). During the various treatments, the cells in the DOX group showed 5503 significantly upregulated genes and 5599 downregulated genes, with 12,574 genes not significantly changed compared to the untreated group. In the upregulated gene pool of the DOX group, 986 and 3914 genes were significantly reversed by DXZ and MgHCF treatment, respectively. In the downregulated genetic pool, the reversed gene numbers were 1528 and 3329, respectively, by DXZ and MgHCF coincubation (Fig. 4g). To better compare the cardioprotective capabilities of DXZ and MgHCF, we defined four gene percentages based on the cascade regulations

and comparisons between DOX vs. Unt and DXZ (MgHCF) vs. DOX, which included pathology-reversed regulation percentages by DXZ and MgHCF and pathology-associated percentages of the same and nonsignificant regulation by DXZ or MgHCF. In addition, the pathologically irrelevant percentages indicated nonsignificant regulation by both DOX and DXZ (or MgHCF), leaving the last gene percentages self-regulated by DXZ or MgHCF (Supplementary Fig. 11). After mapping, we found that MgHCF significantly reversed 30.59% of the DOX-induced pathological genes, in comparison to the 10.62% of the genes reversed by DXZ treatment, demonstrating the potent cardioprotective genetic regulation effect of MgHCF. In addition, lower pathology-associated (16.30%) and pathology-irrelevant (39.72%) percentages were obtained in the D_MgHCF group than those in the D_DXZ group (36.27% and 51.96%, respectively) indicating that MgHCF coincubation effectively reversed the cytotoxicity of DOX-treated cardiomyoblasts at both the cellular and transcriptional

levels and outperformed the clinically applied cardioprotective agent DXZ (Fig. 4i, j).

## MgHCF ameliorates cardiotoxicity by iron binding and antioxidation

We then investigated DOX pathology against cardiomyoblasts on transcriptional, proteomic, and molecular bases in detail. Among the DEGs in the DOX-treated cardiomyoblasts, approximately 92 KEGG pathways (*p*-value cut-off = 0.05) were significantly enriched. From these pathways, treating cardiomyoblasts with DOX-induced significant numbers of phenotypes associated with or very similar to those observed in Alzheimer's disease (AD), Huntington's disease (HD) and Parkinson's disease (PD), with gene ratios of 0.0678, 0.061 and 0.053, respectively, and the significance being the highest among all analysed items (Fig. 5a). Significantly enriched genes also included those involved in apoptosis (*$P$ = 0.047), ferroptosis (*$P$ = 0.033) and autophagy (**$P$ = 0.002), revealing the combinational and comprehensive cell death fate by the pathological changes[34]. By comparing the genetic intersections between the pathological pathways (AD, HD and PD) and cell death pathways (apoptosis, ferroptosis, autophagy and mitophagy), we found that the intersecting genes were most commonly associated with the apoptosis pathway, as shown in the Venn diagrams (Supplementary Figs. 12, 13). Focusing on the apoptosis and ferroptosis pathways, we next compared the fold changes in gene expression among the different treatment groups (Fig. 5b, Supplementary Figs. 14). Cells challenged with DOX showed upregulation of *Parp1*, *Cycs*, *Bax*, and *Casp12* and downregulation of *Bcl2l1*, *Apaf1* and *Bcl2l11*, indicating cell apoptosis. Minor fold changes in these genes were observed after DXZ treatment, demonstrating the non-reversal of apoptotic cell fate. In contrast, apoptosis-associated gene expression was very much ameliorated (Fig. 5b) in the MgHCF treatment group (D_MgHCF). Noticeably, the significance values for apoptosis enrichment changed remarkably from 0.047 (*$P$ for DOX vs. Unt) to 0.074 (nonsignificant, D_MgHCF vs. Unt) and to 0.005 (**$P$ for D_DXZ vs. Unt), validating that apoptotic cell fate can be reversed most effectively by MgHCF and moderately by DXZ as analysed at the transcriptional level (Supplementary Fig. 15). It was also observed that DOX-challenged cardiomyoblasts experienced typical apoptosis at the protein expression level, as indicated by the enhanced expression of apoptosis regulators, such as Aif and Bax, cleaved Casp3, cleaved Parp1 and Cycs, in the DOX group and the D_DXZ group. The much lower expression levels of these proteins and the ameliorated expression levels of Bid and Bcl-xl in the D_MgHCF group revealed the cardioprotective effect of MgHCF at the proteomic level (Fig. 5c, d, Supplementary Fig. 16).

At the molecular level, by staining the cardiomyoblasts that received different treatments with the ROS-sensitive probe DCFH-DA, it was found that DOX-induced abundant production of reactive oxygen species within the cardiomyoblasts by oxidative catalysis, leading to significant oxidative stress within the cells (Supplementary Fig. 17). To our surprise, coincubation of DOX-treated cardiomyocytes with MgHCF ([Fe] = 40 μg ml⁻¹) and DXZ (50 μg ml⁻¹) significantly relieved the oxidative stress inside the cardiomyoblasts (Fig. 5e, Supplementary Figs. 18, 19). More importantly, abnormal intracellular ferrous ion levels were also confirmed by intracellular FerroOrange dye-stained H9c2 cells treated with DOX. Untreated cardiomyoblasts displayed weak intracellular orange fluorescence, implying a normal intracellular ferrous ion level. Intense fluorescence signals were seen in the DOX-treated cardiomyoblasts, and a compromised morphology of the cell nuclei was also observed (Supplementary Fig. 20). Coincubation with MgHCF NCs effectively reduced the intracellular ferrous ion concentration while preventing cardiomyoblast damage by DOX (Fig. 5f, Supplementary Figs. 21, 22). Additionally, the ROS and FerroOrange profiles were further tested in the human cardiomyocyte cell line AC16. The same conclusion was drawn for the AC16 cardiomyocytes and H9c2 cardiomyoblasts (Supplementary Figs. 23, 24). To quantitatively determine the intracellular iron concentration in the cells, we further employed the ICP method to detect the intracellular iron concentrations in AC16 cardiomyocytes subjected to different treatments. Cells in the control, DOX, D_DXZ, and D_MgHCF groups were harvested, washed, counted and homogenized for elemental detection. We determined the iron concentrations based on the number of cells (per 10⁶ cells) in each group. DOX treatment specifically increased the intracellular iron concentration from 0.099 ± 0.01 to 0.780 ± 0.07 μg/ million cells. Therapeutic treatment with DXZ or MgHCF NCs effectively lowered the intracellular iron concentration to 0.261 ± 0.02 μg/ million cells and 0.008 ± 0.01 μg/million cells, respectively (Supplementary Fig. 25). These experiments further validated that MgHCF NCs could effectively rescue DOX-challenged cardiotoxicity while having high biocompatibility in both cardiomyoblasts and cardiomyocytes.

Furthermore, the transcriptional results showed abnormal iron homoeostasis upon DOX treatment. Fundamental intracellular iron homeostasis is divided into three types. Iron, encapsulated by transferrin, binds to the transferrin receptor for the iron influx. Ferric species can be reduced to ferrous ions upon receiving specific signals, which catalyzes the harmful Fenton reaction to generate oxidative hydroxyl radicals. Ferroportin, encoded by *slc40a1*, is the only intracellular iron channel for iron efflux (Fig. 5g)[35]. Upregulations of *Tfrc* and *Fth1* and downregulation of *Slc40a1* have been found in DOX-challenged cardiomyoblasts, which could eventually lead to the increased intracellular iron influx and decreased iron efflux[36]. Pathologically abnormal iron overload inside cardiomyoblasts is therefore mitigated, accompanied by the regulation of a series of ferroptosis-associated genes (Supplementary Figs. 26, 5h). The changed iron influx/efflux was observed from the western blotting results for Ho-1, Slc40a1, Tfrc, Nrf2 and Fth1, which showed that the pathological ferrous ion traffic in DOX-treated cardiomyoblasts can be largely ameliorated by MgHCF NCs (D_MgHCF) and less significantly ameliorated by DXZ (Fig. 5i, Supplementary Fig. 27).

## In vivo cardioprotective effect of DOX-challenged mice

Based on the in vitro cellular performance and the transcriptional and proteomic regulation of the cardioprotective effects of MgHCF and DXZ, their in vivo biocompatibility and cardioprotective effects were evaluated. We initially administered MgHCF (4.8 mg kg⁻¹) and DXZ (25 mg kg⁻¹) intraperitoneally to healthy C57BL/6J mice for 6 consecutive days. During the 14-day biocompatibility evaluations, all mice were healthy compared to the untreated group. At the end of the evaluation process, the mice were sacrificed, and their major organs were dissected for histopathological analyses. As revealed in the microscopic images of the haematoxylin and eosin (H&E)-stained tissue sections, no significant pathologies were observed, preliminarily suggesting the good biocompatibility of both MgHCF and DXZ at their respective doses to mice (Supplementary Fig. 28). In addition, normal haematological indices were obtained, also suggesting the acceptable systemic biocompatibility of free MgHCF and DXZ (Supplementary Fig. 29). To imitate the multiple dosing scheme during cardioprotection, we also evaluated the biocompatibility of MgHCF and DXZ administration over 6 consecutive days. The mouse body weight profile, blood biochemical parameters and histological examination results collectively revealed the good biosafety of both MgHCF and DXZ after multiple administrations (Supplementary Fig. 30). We also did not observe long-term side effects of the MgHCF NCs by measuring routine blood indices (including red blood cells and haemoglobin) in the one-month evaluation period (Supplementary Fig. 31).

To create a pathological state, a single intraperitoneal injection of 20 mg kg⁻¹ DOX was employed to challenge the healthy C57BL/6J mice, which were then treated in combination with the MgHCF NCs prophylactically (once per day for three days before and after DOX challenge, 4.8 mg kg⁻¹) and antidotally (once per day for 6 consecutive days

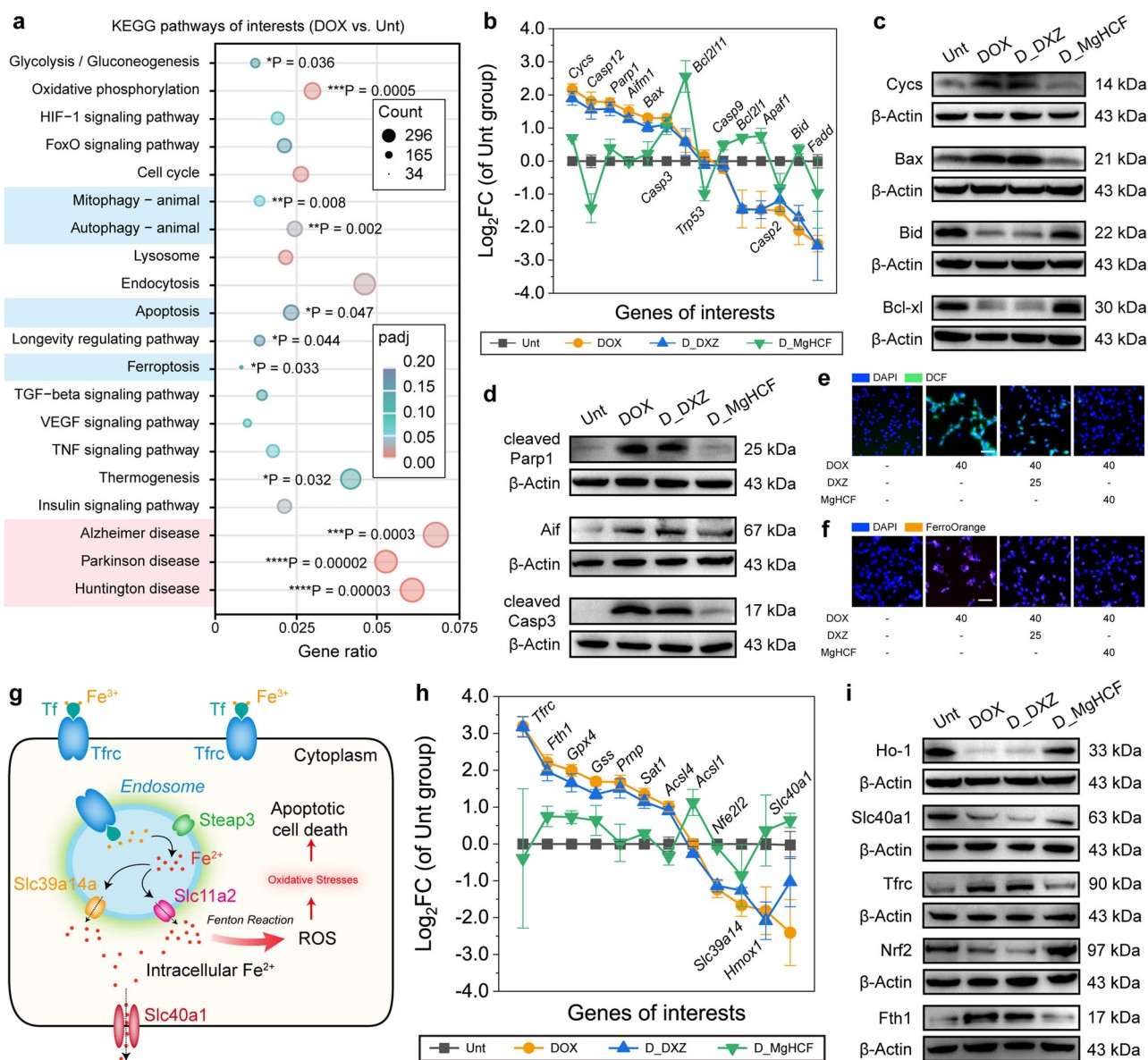

**Fig. 5 | In vitro therapeutic mechanism exploration of cardiac protection.**
**a** KEGG enrichment pathway for DOX-treated cells as compared to the untreated cells. Significant tests (padj) are based on the negative binomial distribution with further corrections (Benjamini–Hochberg procedure). *$P$ < 0.05, **$P$ < 0.01, ***$P$ < 0.001 and ****$P$ < 0.0001. **b** Distributions of the cell-apoptosis-associated genes in different groups. $n$ = 3. Data are presented as mean ± s.d. **c** and **d** Protein expressions of Cycs, Bax, Bcl-xl (**c**), cleaved-Parp1, Aif and cleaved-caspase-3 (**d**)

within cells treated in different groups. **e** and **f** Confocal microscopic images of DCFH-DA-stained (**e**) or FerroOrange-stained (**f**) H9c2 cells received several treatments indicated. Scale bar: 50 μm. **g** Schematic illustrations of the intracellular iron overload, which will lead to oxidative stress and cell apoptosis. **h** Distributions of the iron-transport-associated genes in different groups. $n$ = 3. Data are presented as mean ± s.d. **i** Protein expressions for Ho-1, Slc40a1, Tfrc, Nrf2 and Fth1 within cells treated in different groups.

after DOX challenge, 4.8 mg kg$^{-1}$ (Fig. 6a). Mouse body weight and survival profiles were monitored for the evaluation timeframe of 18 days (Fig. 6b, c). Compared to the normal growth of the untreated mice (control group), mice receiving DOX challenge on Day 1 (DOX group and DOX + MgHCF group) suffered substantial weight loss over the next three days. Four of the mice in the DOX group ($n$ = 6) died within 5 days of the DOX challenge, while the remaining two mice were feeble. The health of the mice in the DOX + MgHCF group was, however, much ameliorated after consecutive MgHCF antidote treatments. These mice (DOX + MgHCF group) showed body weight recovery to a certain extent throughout the remainder of the evaluation period without death, indicating significant relief of the systemic damage caused by DOX and improved health conditions produced by the MgHCF antidote. The changes in body weights of the mice that received prophylactic MgHCF treatment were not significant

compared to that of the control group, revealing the potent cardio-protective effect of MgHCF (Fig. 6b, c). During the evaluation, mice from each group were sacrificed to reveal pathological abnormalities as well as blood biochemical indices. H&E staining of the heart tissue sections revealed no pathological abnormalities within the cardio-myocytes, while trichrome staining presented a fibrotic area, indicating tissue damage (6.62% stained blue) in the heart sections of the sacrificed mice in the DOX group (DOX/sacrifice) (Fig. 6d). The fibrotic zones were observed to become larger (15.81%) during the autopsy inspection of the dead mice in the group (DOX/dead). In the MgHCF prophylactic therapy group, only slight fibrosis of the vascular system (3.35%) was seen. Together with antidotic MgHCF treatment (2.51% of the fibrotic zone), the pathological fibrotic damage to the cardio-myocytes induced by DOX was reduced to a great extent (Fig. 6d, e). In addition, no other pathological abnormalities were observed from the

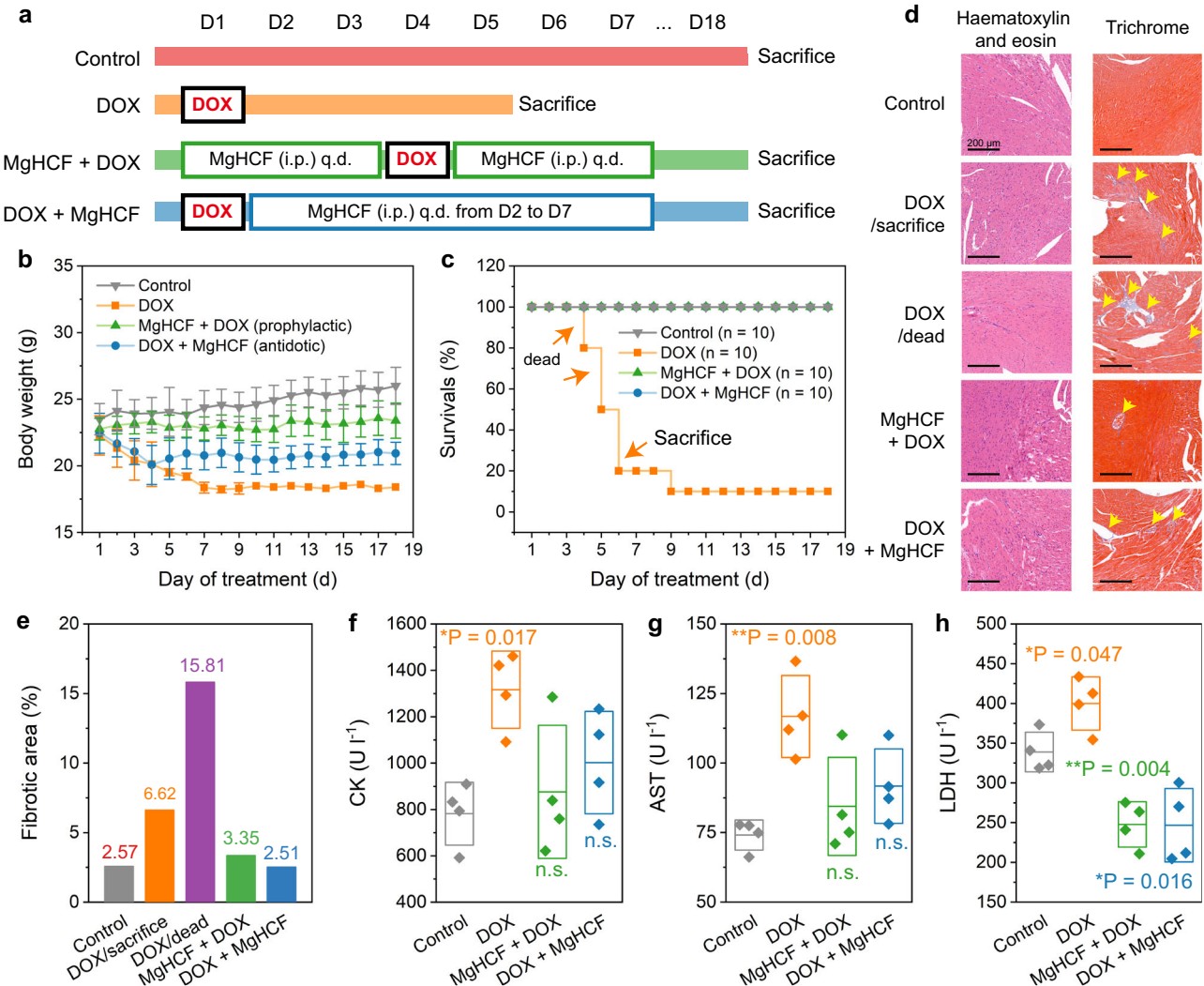

**Fig. 6 | In vivo cardiac function evaluation. a** Dosing schedule of the in vivo cardiac function evaluation. **b, c** Body weights (**b**) and survival rates (**c**) of mice from different groups during the evaluation period. *n* = 10. Data are presented as mean ± s.d. **d** H&E staining and Masson's Trichrome staining images of the heart sections in different groups. **e** Impaired zone percentages on the heart tissue of mice in different groups revealed by the Masson's Trichrome staining imaging. **f–h** Plasma CK (**f**), AST (**g**) and LDH (**h**) levels of each mouse in different groups at the end of the evaluation period. *n* = 4. Data are presented as mean ± s.d. Statistical significance (to the control group) is assessed by Student *t*'s two-tailed test. *$P < 0.05$, **$P < 0.01$ and n.s. for non-significant.

slices of the major organs (i.e., heart, liver, spleen, lung, and kidney) (Supplementary Fig. 32). In addition, creatine kinase (CK) is a critically important biomarker of cardiomyocyte malfunction. Plasma aspartate aminotransferase (AST) and lactate dehydrogenase (LDH) are also sensitive blood biochemical indices that dictate tissue injury[37]. We found that mice challenged with DOX had exceptionally high plasma CK, AST and LDH levels, reflecting impaired tissue functionalities and unhealthy conditions. Prophylactic and antidotic MgHCF therapeutics effectively alleviated these abnormal pathologies, restoring the plasma CK, AST and LDH levels to normal and close to those of the untreated control group (Fig. 6g, h). From the routine haematological indices, both of the MgHCF therapeutics significantly restored the pathological haemogram disrupted by DOX treatment to normal, especially in terms of the indices of white blood cells (WBCs) and lymphocytes (LYMPHs) (Supplementary Fig. 33).

The in vivo cardioprotective effect enabled by the clinically approved drug DXZ was also evaluated through prophylactic and antidotic administration (25 mg kg$^{-1}$) after DOX administration (single dose, 20 mg kg$^{-1}$) to healthy C57BL/6J mice (Supplementary Fig. 34a). During the 14-day evaluation period, a minor body weight decline was observed in the mice in all treatment groups (DOX, DXZ + DOX and

DOX + DXZ groups). One of the mice in the DOX + DXZ group did not survive and died on Day 9. From these weight and survival data, prophylactic DXZ treatment was specifically preferred (Supplementary Fig. 34b, c). Surviving mice were sacrificed at the end of treatment, and their heart tissues and blood were collected for pathological and haematological analyses, respectively. From the H&E-stained heart sections, the arranged cardiomyocytes did not have significant infiltration of monocytes or eosinocytes. No pathological features were observed in the liver, spleen, lung or kidney sections (Supplementary Fig. 35). We also identified the damaged fibrotic tissue area by Masson's trichrome staining of the heart tissues. Establishing DOX pathology induced substantial tissue fibrosis in the mice with an area fibrosis percentage of 26.27% in comparison to that of the non-pathological group (0.79%). Prophylactic DXZ treatment moderately reduced the abnormal fibrotic area to 14.7%, while antidotic DXZ treatment recovered the fibrotic area to only 18.62% (Supplementary Fig. 36). Nevertheless, neither prophylactic nor antidotic treatment could significantly lower the abnormal haematological biochemical indices (i.e., CK, AST and LDH) to normal levels compared to the control group (Supplementary Fig. 37). These results collectively validate the less sufficient cardioprotective performance of DXZ when

employed either prophylactically or antidotally in comparison with the MgHCF therapeutic.

To investigate the in vivo cardiac functionalities, we employed echocardiography (ECG) for periodic inspections of the mice. According to a previous experiment, prophylactic dosing gives significant cardioprotective effects. Hence, in this experiment, mice in the MgHCF + DOX and DXZ + DOX therapeutic groups were pre-treated with MgHCF (4.8 mg kg⁻¹) and DXZ (25 mg kg⁻¹) by intraperitoneal injection for 3 days (q.d.), followed by the induction of DOX pathology with a single i.p. injection of DOX (20 mg kg⁻¹). The present experiments involved two DOX treatments in combination with cardioprotective agent administration within the timeframe of 38 days (Fig. 7a). Pathology was directly induced in the mice in the DOX group on the same day without any pre-treatment. The cardiac functions of these mice were then examined periodically by ECG, including left ventricular internal diastolic diameter (LVIDD), left ventricular internal systolic diameter (LVISD), left ventricular ejection fraction (LVEF) and left ventricular fractional shortening (LVFS). The results of the ECG exam before pre-treatment (ECG Ctrl) and the first ECG exam (ECG I) show that the average LVEF values of the mice in the DOX-treated group dropped out of the floor limit (i.e., 50%) when compared to the untreated group, showing a significant difference. These indices implied a decrease in

blood pumping functionality resulting from the induction of severe DOX pathology (Fig. 7b, c). Mice that received daily MgHCF or DXZ supplementation displayed substantially ameliorated cardiac functions, suggesting that significant cardioprotection was offered by MgHCF, which outperformed the clinical drug DXZ (Fig. 7b, c). DOX was further dosed for another treatment cycle, accompanied or not with the cardioprotective treatments. Judging from the B-mode images (Fig. 7h) and LVEF (Fig. 7b−g) and LVFS values (Supplementary Fig. 38) from the ECG examinations, we found that supplementation of MgHCF NCs as the cardioprotective agent not only significantly improved the cardiac functions of the mice that received DOX challenge but also showed much better protection than the agent clinical DXZ during the examined repetitive dosing and therapeutic cycles.

It has been documented that magnesium overload is potentially harmful to cardiac function. During MgHCF cardioprotection, magnesium ions were released from the nanomedicine, which was accompanied by iron uptake. Such ion exchange occurs in equimolar concentrations. After injection of 4.8 mg kg⁻¹ d⁻¹ MgHCF NCs, the highest Mg²⁺ flux was obtained when all of the magnesium ions were released into the cardiomyocytes (i.e., 4.8 mg kg⁻¹). Therefore, we employed multiple doses of MgHCF NC (4.8 mg kg⁻¹) or MgCl₂ (10 mg kg⁻¹) injections and assessed the biosafety through

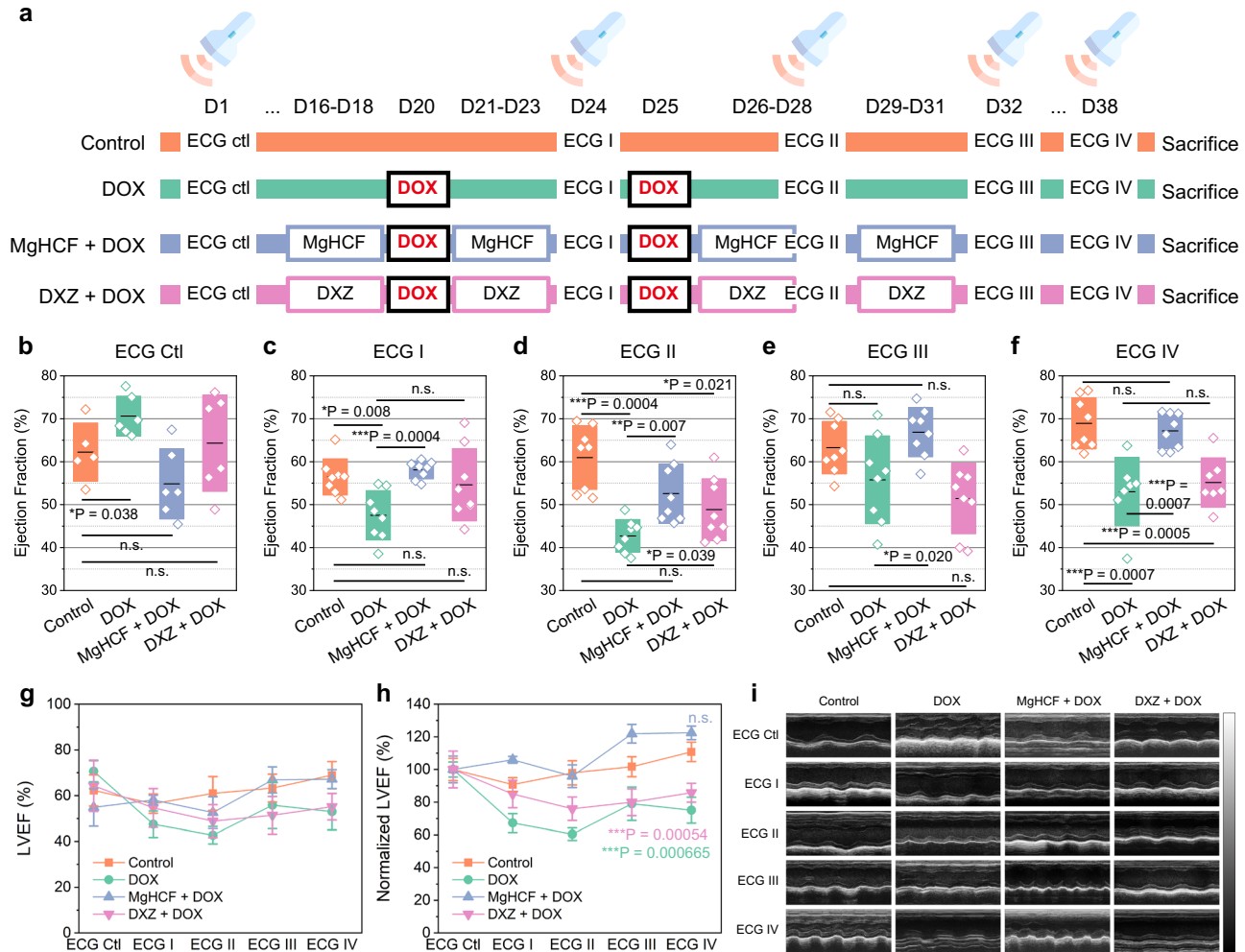

**Fig. 7 | In vivo echocardiography evaluation. a** In vivo dosing schedule for in vivo echocardiography evaluation. **b–f** Left ventricular ejection fractions of mice in different groups by echocardiography inspections: **b** ECG Ctl (n = 6); **c** ECG I (n = 8); **d** ECG II (n = 8); **e** ECG III (n = 8); **f** ECG IV (n = 8). Data are presented as they are and mean (line) ± s.d. (bounds of box). Statistical significance is assessed by Student t′s two-tailed test. *P < 0.05, **P < 0.01, ***P < 0.001 and n.s. for non-significant. **g** and

**h**, Raw (**g**) and normalized LVEF levels (**h**) of mice by different ECG inspections for different groups. Data are presented as mean ± s.d. Statistical significance (to the control group) is assessed by Student t′s two-tailed test. ***P < 0.001 and n.s. for non-significant. **i** Representative echocardiographic images for mice in different groups by echocardiography inspections.

echocardiography. In the present investigation, both MgHCF NCs and $Mg^{2+}$ exhibited good cardiac biocompatibility during the 7-day evaluation timeframe. According to the echocardiographic and electrocardiogram inspections, direct intraperitoneal injection of free $Mg^{2+}$ was safe at an $Mg^{2+}$ dose below 10 mg kg$^{-1}$, and no cardiac toxicity or other abnormalities were observed with the current injection doses of MgHCF NCs (Supplementary Fig. 39). We also concentrated the MgHCF NCs for dose biosafety evaluation. Mice intraperitoneally administered a single typically high dose of 30 mg kg$^{-1}$ MgHCF NCs had an overall survival rate of 80%, while all of the mice administered lower doses of 15, 10 and 7.5 mg kg$^{-1}$ survived for at least two weeks. The median lethal dose of MgHCF NCs for mice was therefore above 52.5 mg kg$^{-1}$ (Supplementary Fig. 40).

### In vivo pharmacokinetics of MgHCF

The pharmacokinetic performance of MgHCF NCs is the most pivotal evaluation for their clinical translation. To initiate this examination, plasma circulation of free $Mg^{2+}$ from the MgHCF NCs was detected before and after i.p. administration of MgHCF NCs at a dose of 4.8 mg kg$^{-1}$. Upon injection, the plasma concentration of $Mg^{2+}$ increased from $9.29 \pm 1.25$ to $26.38 \pm 0.99$ μg ml$^{-1}$ and then gradually decreased to $17.78 \pm 1.18$ μg ml$^{-1}$ at 30 min post-injection and to $8.93 \pm 0.65$ μg ml$^{-1}$ at 4 h post-injection. The data were plotted and fitted exponentially, with the calculated half-life of MgHCF NCs in blood being ~1.59 h (Supplementary Fig. 41a). At $t = 0$, the plasma concentration of $Mg^{2+}$ was determined to be 26.38 μg ml$^{-1}$. For an average mouse with a weight of 20 g and a total blood volume = 2 mL, the drug utilization of MgHCF NCs was then calculated to be 54.96% following the equation:

$$w = C_0/C_{ID} = C_0/(ID * m/V) \times 100\%$$

where $w$ represents the drug utilization, $C_0$ represents the plasma concentration of $Mg^{2+}$ at $t = 0$, ID represents the injection dose, i.e., 4.8 mg kg$^{-1}$, m represents the weight of the mouse, and $V$ represents the total circulating blood volume of the mouse.

For further tissue biodistribution assays, nine C57BL/6J mice were randomly divided into three groups (2, 12 and 24 h) and subcutaneously injected with 4T1 tumour cells for xenograft establishment. The volumes of the xenografts were allowed to grow to 150 mm$^3$ prior to the intraperitoneal administration of a single dose of MgHCF NCs (4.8 mg kg$^{-1}$). At predetermined timepoints (2, 12, 24 h), mice from the corresponding groups were sacrificed for major organ dissection, weighed, homogenized and analysed. From the magnesium distribution profile, we found that upon MgHCF NCs administration, a major distribution of MgHCF NCs was observed in the liver, spleen and lung at 2 h post-injection. At 12 and 24 h post-injection, the overall distribution of $Mg^{2+}$ in these major organs gradually decreased. Specifically, the distribution of $Mg^{2+}$ in the heart was determined to be $7.16 \pm 0.39$, $5.39 \pm 1.77$ and $1.95 \pm 0.27$ ID %/g, respectively. Additionally, $2.84 \pm 0.84$, $1.23 \pm 0.45$ and $1.20 \pm 0.53$ ID %/g had accumulated in the tumours at 2, 12 and 24 h post-injection respectively. These profiles show that upon intraperitoneal administration, the MgHCF NCs were immediately absorbed into the bloodstream and redistributed to the major organs, such as the heart, liver, lung and spleen (Supplementary Fig. 41b). With a half-life of 1.59 h, the MgHCF NCs were gradually eliminated from the body.

In vivo side effects of MgHCF during cardioprotection. Since DOX has been widely employed as an anticancer agent, cardioprotective agents should not affect its antitumour efficacy. To study this, 20 BALB/c mice were randomly divided into four groups (untreated group, DOX group, DOX + MgHCF group and DOX + DXZ group) and subcutaneously injected with 4T1 tumour cells for xenograft establishment. At an average xenograft volume of 150 mm$^3$, a single dose of DOX (20 mg kg$^{-1}$) was intravenously injected into the mice in all

treatment groups to initiate the antitumour evaluations (Fig. 8a). During the next 15 days, strong harmful side effects from DOX were observed in the DOX group, suggesting that DOX-triggered systemic toxicity in the mice (Fig. 8b) while suppressing xenograft growth (Fig. 8c). In comparison, the DOX + MgHCF and DOX + DXZ groups showed significantly stronger suppression of tumour growth than the DOX group (Fig. 8c, Supplementary Fig. 42) in addition to their cardioprotective effects. At the end of the therapeutic evaluation, all of the mice were sacrificed, and the major organs and tumour xenografts were dissected (Fig. 8d). The dissected tumour weights were $2.16 \pm 0.60$, $1.14 \pm 0.20$, $1.09 \pm 0.35$ and $1.42 \pm 0.49$ g (Fig. 8e), while the volumes were $1571 \pm 348$, 995, $838 \pm 203$ and $869 \pm 298$ mm$^3$, respectively, in the control, DOX, DOX + MgHCF and DOX + DXZ groups (Fig. 8f) and statistically indicated that both MgHCF and DXZ would not deteriorate the antitumour chemotherapeutic efficacy of DOX. The ratios of the major organ weights to body weight showed that DOX treatment led to remarkable decreases in these ratios, especially those of the heart (from 0.48% to 0.04%), liver (from 5.70% to 2.86%) and spleen (from 0.73% to 0.05%). These indices were unaffected in the mice that received MgHCF and DXZ treatment following DOX injection (Fig. 8g). Histopathological H&E staining and TUNEL and antigen-Ki67 expression immunofluorescence staining of ultrathin tumour sections from different groups revealed comprehensive cell apoptosis and tissue damage by DOX (Fig. 8h, i, Supplementary Fig. 43). We also assayed the three typical plasma cardiac function biomarkers (AST, CK, LDH) and found that MgHCF substantially lowered the serum AST, CK and LDH levels that were initially upregulated by DOX. DXZ treatment rescued these key biomarkers moderately or even ineffectively compared to the DOX group (Fig. 8j).

In addition, it has been documented that the clinical chemoprotective drug DXZ may induce secondary myelosuppression side effects during application[38]. From the plasma biochemical evaluation (Fig. 8k), it was observed that DOX treatment specifically led to significant reductions in the populations of WBCs and LYMPHs. MgHCF protection effectively recovered the LYMPH population, while the WBC population was also restored to a certain extent. Of note, the DOX + DXZ group showed extensive reductions in the eosinophil (EO), LYMPH and WBC cell populations, evidencing the myelosuppressive effect of DXZ. Additionally, DXZ also caused significant reductions in the total population of platelets (PLTs) and red blood cells (Fig. 8l). To summarize, both DOX and its clinically used protective agent DXZ induced secondary myelosuppressive effects. In comparison, MgHCF significantly mitigated both side effects, cardiotoxicity and myelosuppression, induced by the anticancer drug DOX. MgHCF is therefore considered to have great potential as a substitute for the clinical cardioprotective agent DXZ during anticancer treatment with DOX.

## Discussion

Chemotherapy has been one of the most common modalities to treat cancer ever since its emergence in the 1940s. Although chemotherapy has been widely employed against a broad spectrum of cancers and has produced effective inhibition, growing concerns have been raised due to the accompanying cardiotoxic side effects[39,40]. In addition to chemotherapy, patients exposed to ionizing radiation during radiotherapy have also reported dose-dependent cardiac injury due to ischaemia, valvular disorders, or pericardial injury[41]. Hence, evaluations and treatments of cardiotoxic side effects have remained the major hurdle for oncologists and cardiologists during treatment with tumour therapeutics.

Since chemodrug-induced cardiotoxicity is often dose-cumulative, dose limitations, a prolonged infusion treatment schedule, molecular modification of the drug and chemical cardioprotection are four major strategies to attenuate cardiotoxicity in the clinic. Among these, both dose limitation and a prolonged infusion treatment schedule are conserved modalities that may substantially affect anticancer

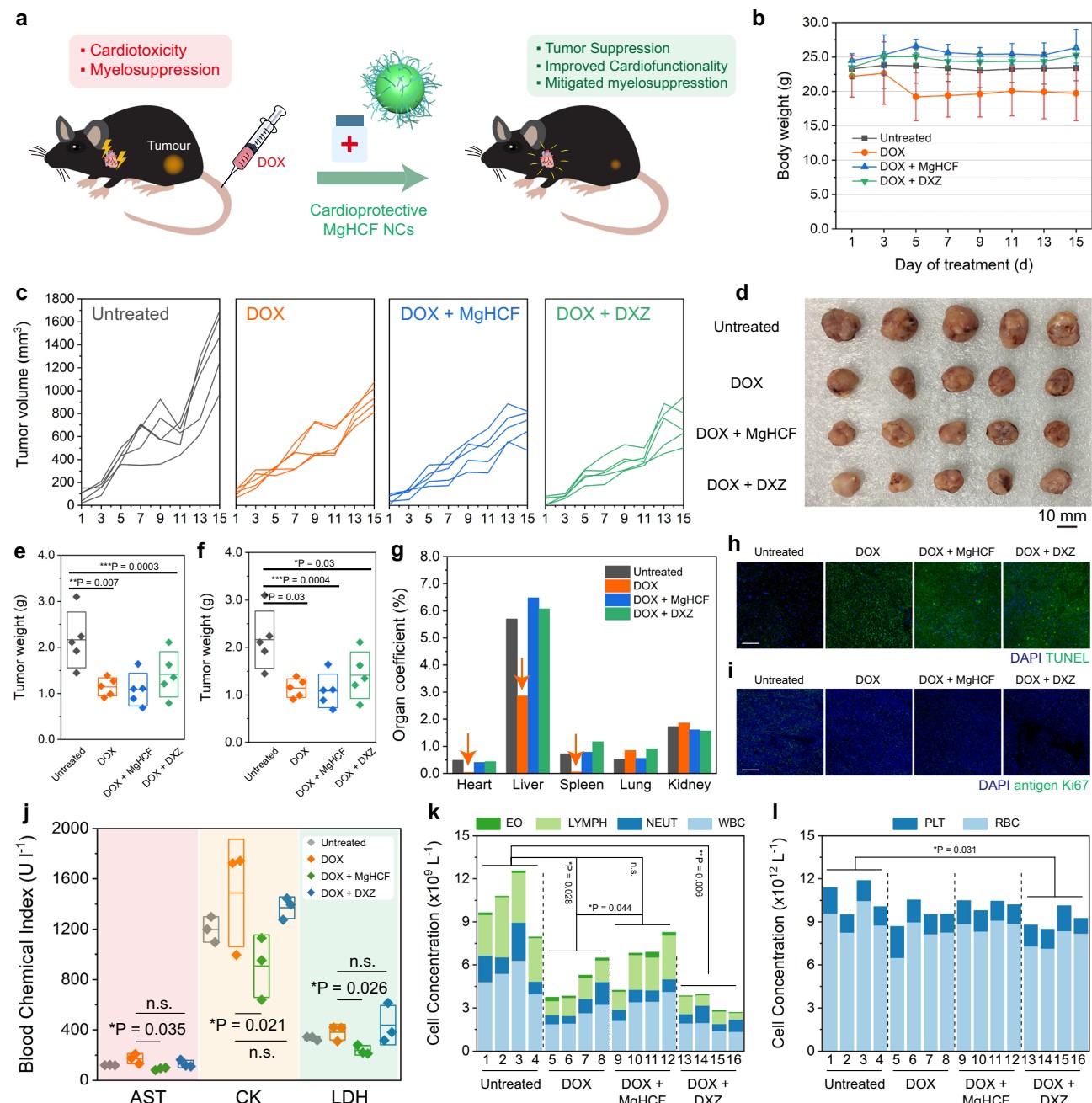

**Fig. 8 | In vivo anti-tumour performances and side effect evaluation.**
**a** Schematic illustration of the in vivo anti-tumour experiment. **b** Body weight profiles of mice in different groups during the in vivo anti-tumour investigation. *n* = 5. Data are presented as mean ± s.d. **c** Xenograft development curves for each mouse from different groups in the therapeutic evaluation period. **d** Digital photograph of the dissected tumour xenografts from each mouse of different groups. **e** and **f** Tumour weights (**e**) and tumour volumes (**f**) of the xenografts dissected from each mouse of different groups. *n* = 5. Data are presented as they are and mean ± s.d. Statistical significance (to the control group) is assessed by Student *t*'s two-tailed test. **P* < 0.05, ***P* < 0.01 and ****P* < 0.001. **g** Organ coefficient of mice from different groups, calculated as the ratio of the organ weight to body weight.

**h** and **i** TUNEL (**h**) and antigen-Ki67 (**i**) immunofluorescence staining images of the ultrathin tumour sections dissected from mice of different groups. **j** Plasma AST, CK and LDH levels of each mouse from different groups. *n* = 3. Data are presented as they are and mean ± s.d. Statistical significance is assessed by Student *t*'s two-tailed test. **P* < 0.05. **k** Cell populations of EO, LYMPH, NEUT and WBC were assayed for each mouse from different groups. Data are presented as they are. Statistical significance is assessed by Student *t*'s two-tailed test. **P* < 0.05, ***P* < 0.01 and n.s. for non-significant. **l** Cell populations of PLT and WBC were assayed for each mouse from different groups. Data are presented as they are. Statistical significance is assessed by Student *t*'s two-tailed test. **P* < 0.05.

---

treatment and induce the risk of tumour metastasis, deteriorating the prognosis of the patient[7]. In addition, cardiotoxicity induced by chemodrugs (e.g., doxorubicin) has been designated as type-1 cardiac damage (direct myocyte death), indicating irreversible myocyte injury after initial chemodrug contamination. It should be noted that intraperitoneal injection has thus been adopted and is one of the most

prevailing systematic administration methods to achieve plasma drug pharmacokinetics similar to intravenous administration[42]. Such high representativeness may benefit future clinical transformation. For rodents such as mice, the cumulative dose of DOX required to induce pathology is approximately 20 mg kg⁻¹ (i.p.). However, for cancer patients receiving DOX chemotherapeutics, multiple scheduled doses

are always necessary. Clinical reports have revealed that the incidence of congestive heart failure was 18% for patients who had received a cumulative DOX dose of 551–600 mg m$^{-2}$. This incidence could further increase to 36% for patients who had received doses higher than 600 mg m$^{-2}$ [11]. Particular care, including electrocardiography, echocardiography and endomyocardial biopsy, should be taken to identify such pathology.

Molecular modification of drugs requires a fundamental understanding of the molecular structure of the chemodrug, followed by more efforts in structural modification as well as a comprehensive clinical reassessment of the biosafety and therapeutic efficacy before drug approval[43]. The development of a chemical cardioprotective agent is the most ideal strategy to attenuate or even reverse cardiotoxicity without affecting the cancer treatment schedule. These cardioprotective agents can always be co-administered with chemodrugs, maximizing the cardioprotective effect. Therefore, the design and development of chemical cardioprotective agents are of high clinical significance.

Chemical cardioprotective agents are designed according to the molecular mechanism of the pathology of the cardiotoxicity generated by chemodrugs or radiation. In the present report, we focused on the fundamentals of DIC for the development and application of cardioprotective agents. According to our cellular mRNA sequencing results, we observed a combination of cell death pathways in DOX-contaminated cardiomyoblasts, such as apoptosis, ferroptosis, autophagy and mitophagy, with apoptosis being the most dominant effect induced by DOX chemotherapy. As the transcriptome results collectively enriched mitochondrial dysfunction, an abnormal ferrous ion traffic-mediated intrinsic apoptosis pathway has also been revealed and identified. Of note, although such transcriptome regulation contributed to the significant enrichment in the ferroptosis pathway by KEGG analysis (Fig. 5), the scored gene ratio (0.779%) remained relatively low (compared to 2.336% for apoptosis). According to the high upregulation of *Gpx4*, which encodes glutathione peroxidase for lipid peroxidation clearance in DOX pathology, the pathological regulation of intracellular iron is believed to be insufficient to cause prominent cell ferroptosis due to the non-destruction of the anti-lipid peroxidation system. In addition to apoptosis, autophagy and mitophagy were also found to be actively involved in the ultimate cell fate of cardiomyoblasts, shedding light on the design of autophagy inhibitors and their applications in cardioprotection in the near future.

Focusing on the molecular mechanism of DIC, requirements for the normalization of cellular iron traffic as well as oxidative stresses have been presented. As a Prussian blue analogue, the synthetic MgHCF nanocatalysts introduced in the present work were demonstrated to alleviate and prevent the cardiotoxicity induced by DOX contamination both in vitro and in vivo. Its ferrous ion chelation and catalytic antioxidation properties were demonstrated to outperform the clinically approved DXZ in terms of cardioprotection. To move a step forward, large-scale synthesis and preclinical large animal studies are encouraged. The preparation of MgHCF nanocatalysts involves the facile procedures of simple mixing, dialysis and freeze-drying. Gram-scale synthesis could be achieved with several synthetic batches. The cardioprotective effects of MgHCF nanocatalysts on large rodents (e.g., rats) and other large animals are worth investigating.

In summary, we synthesized an MgHCF nanocatalyst by a facile PVP-directed self-assembly approach that possesses potent ferrous ion capture and antioxidation functions for cardioprotection during tumour chemotherapy. Based on the excellent superoxide dismutase and H$_2$O$_2$ decomposition catalytic activities to eliminate cytotoxic radical species, the viability of the cardiomyoblasts impaired by DOX was effectively rescued by the MgHCF NCs to a greater extent than the much less effective DXZ agent currently applied in the clinic, which was evidenced and distinguished by regulating apoptosis and ferrous ion traffic at both the mRNA and protein levels. MgHCF NCs also exhibited excellent in vivo cardioprotective effects by attenuating the cardiotoxicity and harmful side effects due to DOX contamination without deteriorating the antitumour effect of the chemodrug. The highly efficient cardioprotective effect and excellent biocompatibility will make these MgHCF nanocatalysts one of the most promising and clinically transformable cardioprotective agents to be employed in future cancer treatments due to their ability to mitigate side effects and protect organs during anticancer chemotherapy.

## Methods

### Synthesis of MgHCF NCs

MgCl$_2$·6H$_2$O (0.8 mmol) and polyvinylpyrrolidone (Mw = 10,000, 300 mg) were completely dissolved into 40 mL of aqueous solution (pH = 1, adjusted by HCl) under vigorous stirring. Then 40 mL of aqueous solution (pH = 1, adjusted by HCl) containing potassium ferricyanide (0.8 mmol) was injected into the above solution at speed of 80 mL/h. The reaction was allowed to proceed at room temperature for 2 h. Afterwards, the mixture was dialyzed against deionized water for 6 h. The obtained dispersion was then concentrated by evaporation in vacuo, followed by freeze-drying to obtain the MgHCF NCs powders.

### Characterization

Transmission electron microscope (TEM) was adopted for morphology observation on JEM-2100F electron microscope operated at an accelerated voltage of 200 kV. Hydrodynamic diameter and zeta potential profile were obtained on Malvern Zetasizer Nanoseries (Nano ZS90) following the dynamic light scattering (DLS) method. Nanoparticle distribution is statistically presented as mean ± s.d. following the normal distribution. XRD patterns were collected on Rigaku D/MAX-2550 V X-ray power diffractometer with Cu Kα radiation ($\lambda$ = 1.5405 Å) scanning with a $2\theta$ ranged from 5° to 75°. Fourier transforms infrared (FT-IR) spectra were acquired on Nicolet iS10 spectrometer using KBr pellets. Raman spectra were collected on Renishaw inVia spectrometer with a laser parameter of 473 nm, 0.2 mW. The XPS spectra were obtained on Thermo Scientific ESCA-LAB 250 spectrometer and the results were calibrated following the 284.8 eV of the C1$s$. UV–vis spectra were assayed on a UV-3100 Shimadzu spectrometer. ICP-OES spectra were obtained on Agilent 700 Series instrument.

### Ferrous binding assay

1 mL of serial concentrations of the aqueous solution containing 40, 20, 10, 5, 2.5, 1.25 and 0 µg mL$^{-1}$ iron(II) ammonium sulfate was added into the aqueous solution of MgHCF NCs (40 µg mL$^{-1}$) in identical volume. The mixture was allowed for 30 s equilibrium. UV–vis absorption spectra were obtained. For ferrous binding assays in the dialysis membrane isolated cation exchange process, 10 mL of the MgHCF NCs ([Mg$^{2+}$] = 20 µg/mL) was sealed into the dialysis bag and immersed into the beaker containing 150 mL solution of iron(II) ammonium sulphate with stirring. At predetermined time points (0, 0.5, 2, 5, 8, 15, 25, 60, 90, 120 and 240 min), aliquots of the solution in the beaker were collected for elemental analysis (Mg, Fe) using ICP-OES.

### ESR assays

The ESR assays were carried out on Bruker EMX ESR spectrometer using 5-tert-Butoxycarbonyl-5-methyl-1-pyrroline-N-oxide (BMPO) as the radical trapper for superoxide anion radical determination. In a typical assay, an aqueous solution (500 µL H$_2$O) containing xanthine (10 mM), xanthine oxidase (0.5 U) and BMPO (5 mM) were

mixed with MgHCF NCs of varying concentrations. An aliquot of the solution was then transferred into the quartz capillary for ESR measurements.

### In vitro cellular experiments

Rat cardiomyoblasts H9c2 cell line was purchased from the Cell Bank, the Committee of Type Culture Collection of the Chinese Academy of Sciences (Serial# GNR 5). Rat cardiomyocytes AC16 cells were purchased from Guangzhou Cellcook Biotechnology Co. Ltd. (Serial# CC4030). These cell lines have not been listed by the international cell line authentication committee as cross-contaminated or mis-identified cell lines (v8.0, 2016) and have passed the conventional tests of cell line quality including morphology identification, iso-enzymes and mycoplasma. Cells were cultured using DMEM medium containing 4 mM L-Glutamine, 4500 mg l$^{-1}$ glucose, 10% fatal bovine serum (FBS), 100 units ml$^{-1}$ streptomycin and 100 units ml$^{-1}$ penicillin.

**Cell viability assays.** H9c2 cells were inoculated into the 96-well plate with a cell density of 8000 cells per well and incubated for 12 h for cell attachment. The medium was then discarded and rinsed with cold PBS three times. Cells were supplemented with fresh medium containing serial doses of DOX chemodrug. For in vitro cytoprotection assays, MgHCF NCs or DXZ was co-supplemented with the DOX chemodrug for cell treatments. After 12 h co-incubation, the medium was discarded and the cells were rinsed with cold PBS three times, followed by the addition of the fresh medium containing 10% cell-counting kit-8 solution. The optical absorption profile at 450 nm was assayed on a microplate reader after 4 h following the protocol.

**mRNA-sequencing experiment.** The total RNA from the H9c2 cells with different treatments (untreated, DOX treated, DOX + MgHCF and DOX + DXZ) for 6 h were harvested using TRIzol (Invitrogen). The mRNA was purified using poly-T oligo-attached magnetic beads and fragmented using divalent cations under elevated temperature NEBNext First Strand Synthesis Reaction Buffer (5X). First-strand cDNA was synthesized using a random hexamer primer and M-MuLV Reverse Transcriptase (RNase H-). Second-strand cDNA synthesis was subsequently performed using DNA Polymerase I and RNase H. Remaining overhangs were converted into blunt ends via exonuclease/polymerase activities. After adenylation of 3′ ends of DNA fragments, NEBNext Adaptor with hairpin loop structure was ligated to prepare for hybridization. The products were then enriched and purified (AMPure XP system) to form the final cDNA library and the library quality was assessed on the Agilent Bioanalyzer 2100 system. The clustering of the index-coded samples was performed on a cBot Cluster Generation System using TruSeq PE Cluster Kit v3-cBot-HS (Illumia) according to the manufacturer's instructions. After cluster generation, the library preparations were sequenced on an Illumina Novaseq platform and 150 bp paired-end reads were generated. The web-based NovoMagic data analysis platform was used for raw data variance modelling and statistical analyses.

**Western blotting.** The proteins from the H9c2 cells with different treatments (untreated, DOX treated, DOX + MgHCF and DOX + DXZ) for 6 h were harvested and resuspended in protein lysis buffer containing proteinase inhibitors (Roch) with sonication. The total protein amount was quantified using a protein assay kit supplemented by Bio-Rad (Hercules, CA, USA). The obtained lysate proteins were separated by SDS−PAGE gels, transferred onto the PVDF membranes and blocked with BSA (5%) for subsequent primary antibody incubation. Afterwards, the membranes were co-incubated with horseradish peroxidase-conjugated secondary anti-mouse/rabbit antibodies. Bands were visualized by ECL Plus Western Blot Detection Kit (Tanon).

**Confocal observations.** H9c2 cells or AC16 cells with different treatments (untreated, DOX treated, DOX + MgHCF and DOX + DXZ) for 6 h were rinsed with cold PBS three times before the cells were stained with DCFH-DA (2′,7′-dichlorodihydrofluorescein diacetate), FerroOrange and DAPI dyes for 15 min in the incubator. Cells were then washed and again immersed with PBS for confocal microscope observation.

### In vivo animal experiments

The in vivo animal experiments were performed according to the guidelines of the Laboratory Animal Ethics Committee of Shanghai Tenth People's Hospital (SHDSYY-2020-Z0026). C57BL/6J mice (5 weeks old, male) were purchased from Shanghai Laboratory Animal Center (SLAC). Mice were housed in the specific pathogen-free (SPF) facility with temperature/humidity control (22 ± 2 °C, 40–70%) and were freely accessible to water and food. The light/dark cycle is 12/12 h. Mice were monitored every day.

**In vivo biocompatibility experiment.** Thirty C57BL/6J mice (5 weeks old) were divided into three groups randomly (Control ($n = 10$), MgHCF ($n = 10$) and DXZ group ($n = 10$)). For respective groups, MgHCF NCs (4.8 mg kg$^{-1}$) and DXZ (25 mg kg$^{-1}$) suspended in the saline were injected intraperitoneally for 6 consecutive days (q.d.), respectively. Mice of each group were sacrificed for haematological analysis and H&E staining of major organ tissues.

**In vivo cardioprotection experiment.** Forty C57BL/6J mice (5 weeks old) were divided into four groups randomly (Control ($n = 10$), DOX ($n = 10$), MgHCF + DOX ($n = 10$) and DOX + MgHCF group ($n = 10$)). To initiate the cardiotoxicity pathology, a single dose of DOX chemodrug (20 mg kg$^{-1}$) was employed to treat the mice in the DOX group and DOX + MgHCF group via tail-vein. For mice in MgHCF + DOX group, DOX chemodrug was administrated on Day 4. For MgHCF cardioprotective effect investigation, MgHCF NCs suspended in the saline were injected intraperitoneally once per day (q.d., 4.8 mg kg$^{-1}$) from Day 1 to Day 4, and from Day 5 to Day 7 for mice in MgHCF + DOX group. The injection schedule for mice in DOX + MgHCF group is from Day 2 to Day 7. Body weight of mice was recorded every day. Dead mice in the DOX group and one random survival mice of each group were sacrificed with their heart tissue dissected for H&E staining and Masson's Trichrome staining to visualise the cardiotoxicity. At the end of the therapeutic evaluation, all mice were sacrificed and the blood samples were subjected to plasma analysis of CK, AST and LDH indexes.

**In vivo cardiac functionality investigation by echocardiography inspection.** Twenty-three C57BL/6J mice (5 weeks old, male) were divided into four groups randomly (Control ($n = 6$), DOX ($n = 5$), MgHCF + DOX ($n = 6$) and DXZ + DOX group ($n = 6$)). All mice were subjected to ECG inspection (ECG ctl) and housed for two weeks. For mice in MgHCF + DOX group, MgHCF injections (i.p., once per day, 4.8 mg kg$^{-1}$) were conducted on days 16–18, 21–23, 26–28, and 29–31. For mice in DXZ + DOX group, DXZ injections (i.p., once per day, 25 mg kg$^{-1}$) were conducted at the same timepoints for MgHCF injections. DOX chemodrug administrations were given on Day 20 and Day 25. ECG inspections were conducted on Day 24 (ECG I), Day 28 (ECG II), Day 32 (ECG III) and Day 38 (ECG IV). ECG inspections were performed on a Vevo LAZR imaging system (FUJI film VisualSonics) equipped with a 30 MHz transducer. Conventional 2D images and M-Mode images were obtained at the level of midpapillary muscle. LVIDS and LVIDD parameters were measured at M-mode and the LVEF and LVFS were

calculated automatically following the formula (1)–(4):

$$LVVD = \frac{7}{2.4 + LVIDD} \times LVIDD^3 \qquad (1)$$

$$LVVS = \frac{7}{2.4 + LVIDS} \times LVIDS^3 \qquad (2)$$

$$EF\% = \frac{LVVD - LVVS}{LVVD} \times 100 \qquad (3)$$

$$FS\% = \frac{LVIDD - LVIDS}{LVIDD} \times 100 \qquad (4)$$

**Masson's trichrome staining.** Paraffin-embedded tissue sections are initially deparaffinized and rehydrated to distilled water. Slides are then rinsed with potassium dichromate overnight followed by gentle rinses using distilled water. Iron Hematoxylin working solution is used to stain the slides for 10 min, with subsequent rinsing with distilled water. Then the slides are stained with ponceau-acid fuchsin staining solution for 5 min, with subsequent rinsing with distilled water. The slides are then placed into freshly prepared phosphomolybdic acid solution for 10 min, followed by the staining of Aniline Blue solution for 5 min. The slides are finally dehydrated and sealed for microscopic inspections. Images are quantified using ImageJ software with the threshold colour selection mode.

**In vivo anti-tumour experiment.** Twenty C57BL/6J mice (5 weeks old, male) were divided into four groups randomly (Control ($n = 5$), DOX ($n = 5$), MgHCF + DOX ($n = 5$) and DXZ + DOX group ($n = 5$)). 4T1 tumour xenograft models were established by subcutaneously injecting suspended 4T1 tumour cells into the right flank of the mice with a density of $10^6$ cells per mouse. Xenografts were allowed to grow to ~150 mm$^3$ prior to the in vivo anti-tumour experiment. For mice in DOX, MgHCF + DOX and DOX + DXZ groups, mice were intravenously injected with DOX chemodrugs (20 mg kg$^{-1}$). For mice in DOX + MgHCF and DOX + DXZ groups, daily administrations (i.p., q.d.) of MgHCF NCs (4.8 mg kg$^{-1}$) or DXZ (25 mg kg$^{-1}$) were supplemented for 4 days. During the evaluation period, xenograft dimension profiles were measured with a digital calliper perpendicularly ($l_1$, $l_2$, $l_1 > l_2$) and mice body weight profiles were obtained every other day. Xenograft volume was calculated as follows:

$$V = \frac{1}{2} \times l_1 \cdot l_2^2 \qquad (5)$$

At the end of the evaluation, mice were sacrificed by painless cervical dislocation. Tumour xenografts and major organs (heart, liver, spleen, lung and kidney) of mice were harvested for ultrathin section preparations and subsequent histopathology analysis by H&E, antigen Ki67 and TUNEL, respectively. Organ coefficients were calculated as the percentages of the specific organs of the body weight. Plasma routine indexes and plasma biochemical indexes were assayed from the blood samples from the eyeball before cervical dislocation.

**Ethics statement**
The in vivo animal experiments were performed according to the guidelines of the Laboratory Animal Ethics Committee of Shanghai Tenth People's Hospital (SHDSYY-2020-Z0026). The committee agrees to the following humane endpoints: (1) 30% weight loss; (2) tumour burden exceeds 1800 mm$^3$ or one of any dimensions to exceed 18 mm. Mice were monitored every day. For euthanasia, mice were always euthanatized following the painless cervical dislocation.

**Statistical analyses**
The data significance is analysed following the Student's *t*-test: *$P < 0.05$, **$P < 0.01$, ***$P < 0.001$ and ****$P < 0.0001$. Corresponding $P$ values were always indicated. n.s. for non-significant.

**Reporting summary**
Further information on research design is available in the Nature Portfolio Reporting Summary linked to this article.

## Data availability
The mRNA-seq data have been deposited in the Gene Expression Omnibus database with an accession number GSE213983. All data are available in the article, Supplementary Information, and the Source Data file provided with the article. Source data are provided with this paper.

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

## Acknowledgements

The authors acknowledge the financial support from the National Natural Science Foundation of China (Grant nos. 22005327 (M.H.), 21835007 (J.S.), 82001944 (L.W.), 82071004 (P.G.), 51722211 (Y.C.) and 51672303 (Y.C.)), Fundamental Research Funds for the Central Universities (22120220271 (M.H.), 22120220209 (J.S.)), Key Research Programme of Frontier Sciences, Chinese Academy of Sciences (Grant No. ZDBS-LY-SLH029 (J.S.)), Basic Research Programme of Shanghai Municipal Government (Grant No. 21JC1406000 (J.S.)), Research Unit of Nanocatalytic Medicine in Specific Therapy for Serious Disease, Chinese Academy of Medical Sciences (Grant No. 2021RU012 (J.S.)), Programme of Shanghai Academic Research Leader (Grant no. 18XD1404300 (Y.C.)), Shanghai Municipal Government S&T Project (Grant Nos. 17JC1404701 (J.S.), 19JC1415503 (P.G.)) and China Post-doctoral Science Foundation (Grant nos. BX20200345 (M.H.), 2020M671243 (M.H.) and 2019TQ0231 (L.W.)).

## Author contributions

Conceptualization: M.H., Z.T., J.S. and P.G.; MgHCF NCs synthesis and characterization: M.H., Z.T. and L.W.; Technical supports for material characterizations and echocardiographic inspections: L.Z. and H.G.; Investigations: M.H. and Z.T.; Methodology and resources: Y.C.; Writing-original draft: M.H.; Writing-review, and editing: J.S. (lead) and all other authors (supporting); Supervision: P.G. and J.S. All authors read and approved the manuscript.

## Competing interests

The authors declare no competing interests.

## Additional information

[1]Shanghai Tenth People's Hospital, Shanghai Frontiers Science Center of Nanocatalytic Medicine, School of Medicine, Tongji University, 200072 Shanghai, P.
R. China. [2]State Key Laboratory of High Performance Ceramics and Superfine Microstructure, Shanghai Institute of Ceramics Chinese Academy of Sciences;
Research Unit of Nanocatalytic Medicine in Specific Therapy for Serious Disease, Chinese Academy of Medical Sciences (2021RU012), 200050 Shanghai, P. R.
China. [3]Centre of Materials Science and Optoelectronics Engineering, University of Chinese Academy of Sciences, 100049 Beijing, P. R. China. [4]Department
of Ophthalmology, Shanghai Ninth People's Hospital, Shanghai Jiao Tong University School of Medicine, 200011 Shanghai, P.R. China. [5]Shanghai Key
Laboratory of Orbital Diseases and Ocular Oncology, 200011 Shanghai, P.R. China. [6]Department of Ultrasound, The First Affiliated Hospital of Zhengzhou
University, 450052 Zhengzhou, P. R. China. [7]Materdicine Lab, School of Life Sciences, Shanghai University, 200444 Shanghai, P.R. China. [8]These authors
contributed equally: Minfeng Huo, Zhimin Tang. ✉e-mail: guping2009@sjtu.edu.cn; jlshi@mail.sic.ac.cn

