## [Peer Review File · Nature Communications]

Magnesium Hexacyanoferrate Nanocatalyst Attenuates Chemodrug-Induced Cardiotoxicity through Ferromodulation-Driven Anti-ApoptosisREVIEWER COMMENTS

Reviewer #1 (Remarks to the Author):

In this paper, MgHCF nanocatalysts (NCs) have been synthesized through a PVP-directed self-assembly method. The NCs possess potent ferrous capturing and antioxidation functions for cardioprotection during tumor chemotherapy. The viability of the cardiomyocytes impaired by DOX could be effectively protected by the MgHCF NCs. The MgHCF NCs also exhibit excellent *in vivo* cardioprotection effect via attenuating the cardiotoxicity and the harmful side-effect due to the DOX contamination without deteriorating the anti-tumor effect of the chemodrug. The MgHCF NCs is novel and effective for the cardioprotection application. Their *in vivo* and *in vitro* effects have been thoroughly demonstrated. The paper is well organized. But the characterizations of MgHCF NCs are insufficient. This paper can be published after the following revisions:

- (1) In Scheme 1 and Figure S1, what do the green spheres and light blue thorny particles mean? They should be indicated in the figures.
- (2) The detailed structure of MgHCF NCs is not clearly clarified. Actually, the schematic structure in Figure 1a is not supported by the data. More evidence should be supplied to clarify the structure of MgHCF NCs.
- (3) In Figure 1d and 1e, the XRD patterns and FTIR spectra of pure PVP and MgHCF NCs should be supplied for better comparison.
- (4) The ferrous capturing ability of MgHCF NCs should be proved under the interference of intracellular biomacromolecules. Thus the Mg²⁺ and Fe²⁺ exchange experiments should be performed in the presence of cell lysis to mimic the *in vivo* environments.
- (5) After ferrous capturing, how do the antioxidation performances of MgHCF NCs change?
- (6) In Figure S14, why do DCFH-DA-stained cardiomyocytes treated with DOX supplemented with MgHCF at the concentration of 80 ppm show much stronger fluorescence than that treated at the concentration of 40 ppm? And in Figure S15, DCFH-DA-stained cardiomyocytes treated with DOX supplemented with DXZ at the concentration of 25 ppm still showed much stronger green fluorescence, it cannot say that the oxidative stress is significantly relieved.
- (7) Beside the FerroOrange staining method, could the authors supply more quantitative data to measure the intracellular ferrous concentration?

Reviewer #2 (Remarks to the Author):

Hou et al investigated the Magnesium Hexacyanoferrate Nanocatalyst attenuates chemodrug-Induced cardiotoxicity through ferromodulation-driven anti-apoptosis. Excellent *in vitro* and *in vivo* cardioprotection performances of MgHCF NCs have been demonstrated and the underlying intracellular ferrous traffic regulation mechanism has been explored in detail. The marked cardioprotective effect and biocompatibility render MgHCF NCs to be a highly promising and clinically transformable cardioprotective agent to be employed during cancer treatments. This is an interesting finding in the Dox induced cardiotoxicity treatment. However there are some questions need to be explained as follow:

Major:

1. The studies on the biological properties of MgHCF NCs is not sufficient, including the half-life of the drug in the blood, drug utilization, median lethal dose, and drug distribution in different organs after intraperitoneal injection of MgHCF NCs, especially the drug concentration in the heart and tumor should provided.
2. Long-term side effects of MgHCF NCs on the number of red blood cells and hemoglobin should be investigated?
3. The important question in this study is, why the authors didn't study cardiomyocyte ferroptosis, but apoptosis? There have been many reports about DOX induced myocardial ferroptosis. If MgHCF NCs can inhibit iron ions, the main target should be ferroptosis.
4. The H9c2 cells used in this study are derived from myoblast cells, which are closer to those of skeletal muscle cells. H9c2 cells have the ability to proliferate, which is significantly different from

cardiomyocytes. So it is recommended that the author repeat the relevant experiments with primary cardiomyocytes.

Minor:

1. The quality of Fig 4J Western blot is too poor.
2. Fig 6J LVEF% should not use normalized data, but should use raw data.
3. The caspase-3 in Figure is incorrectly labeled, it should be cleaved caspase-3.
4. In the in vivo data, n=4, which does not meet the statistical requirements.
5. The animal survival curve is that n=6-5 does not meet the statistical requirements. Generally, the number of animals in each group should be 10-20.
6. The detection data on autophagy, mitophagy, apoptosis and ferroptosis of cells and animals are insufficient.

In short, this study is interesting, but not sufficient to be published in NC. More functional studies and pharmacological studies data should be provided to support the

Reviewer #3 (Remarks to the Author):

Key results

In cancer patients, treatment with anthracyclines (i.e., doxorubicin) is very often jeopardized by drug-induced cardiotoxicities mostly caused by excessive cellular oxidative stress. Here, the authors clearly demonstrate that the accumulation of radical species is catalyzed by Fenton-like reactions consecutive to alterations in iron transport and report on the development of a new nanomedicine, called MgHCF NCs, specifically designed to capture iron and hence reduce the iron-associated risk. After demonstrating its efficacy in eliminating the cytotoxic radical species in vitro, the authors showed the biocompatibility and cardioprotection of MgHCF NCs in vitro (H9C2 cardiomyoblasts) and in vivo (mice). Finally, in a tumor mouse model, the authors demonstrate that MgHCF NCs do not impair the anticancer efficacy of doxorubicin while significantly reducing its cardiac side effects. Based on convincing data, they conclude that MgHCF NCs displays all properties for a promising cardioprotective agent during cancer treatments.

Validity

The general framework of the research is based on a proven clinical reality according to which patients exposed to anthracyclines, such as doxorubicin, develop cardiotoxic damage during treatment, compromising the chances of success. This adverse effect is caused by the generation of reactive oxygen species, creating oxidative stress induced by the anticancer agent. The phenomenon would be amplified by an alteration of iron transport in heart cells, creating an environment conducive to Fenton reactions catalyzing the genesis of free radicals. This hypothesis has been partially confirmed in previous studies showing the partially protective effect of an iron chelator, dexrazoxane, the only cardioprotective agent used clinically to date in this type of chemotherapy.

In this publication, the authors present convincing results obtained in various in vitro and in vivo models showing the superior efficacy to dexrazoxane of a new nanomaterial, MgHCF NCs. The data clearly show that this nanocatalyst induces excellent cardioprotection thanks to its iron capture (by replacement of initially trapped magnesium ions) and antioxidant properties. In vitro, according to their transcriptomic and proteomic data acquired in the H9C2 line of cardiomyoblasts, the authors show that this nanocatalyst significantly increases the chances of survival of cardiac cells exposed to doxorubicin by reducing both deregulation of iron trafficking and cell apoptosis. In vivo (mice), this cardioprotective effect was partially confirmed by cardiac echocardiography studies. Finally, the authors presented in vivo data in mice showing the verified biocompatibility of the new agent, during repeated exposures over several weeks and also that the latter does not interfere with the anticancer capacities of doxorubicin in a mouse model implanted with a subcutaneous tumor. The encouraging results obtained for the new nanomaterial in terms of cardioprotection are supported by the fact that in the majority of in vitro and in vivo tests evaluated in this study, the latter performed better than dexrazoxane, used here as a positive benchmark.

Overall, the authors' conclusions (claiming that their new MgHCF Ncs nanomaterial is one of the most promising cardioprotective agents for clinical use) are appropriately supported by the data, well

justified and reliable.

Significance

To the best of my knowledge, the authors' working hypotheses on the cellular mechanism leading to doxorubicin-induced cardiotoxicity, as well as their conclusions regarding the efficacy of their new nanocatalyst as a promising cardioprotective agent during cancer therapy are correct. I am not aware of any publication supporting contradictory data.

Data and methodology

First, I would like to mention that I do not have the expertise concerning the physicochemical aspects of characterization of the nanocatalyst at the center of this publication. I will therefore not comment on this part of the results. On the other hand, I feel comfortable with the evaluation of in vitro and in vivo tests of biocompatibility and efficacy of the product. Here are my main comments on this part :

1. Concerning the in vitro evaluation of MgHCF NCs antioxidant properties, I am not convinced with the use of "multi-enzymatic catalytic performance" (see line 196) as well as with the "SOD superoxide dismutase- and catalase-like catalytic activities" (see for example lines 30) of MgHCF NCs. Such wording should be avoided since the agent can indeed induce those effects but lacks enzymatic activity.

2. In the evaluation of MgHCF NCS' cardioprotective properties, the authors used both in vitro and in vivo assays. In the in vitro cellular experiments, they selected the H9C2 cell line and inappropriately called them "cardiomyocytes" (line 739). In fact, H9C2 cells are cardiomyoblasts which exhibit most of the phenotypic characteristics of mature heart cells except the contractile properties. As a result, they are less dependent on oxidative phosphorylation and mitochondrial activity, the targets of doxorubicin studied here. It would be interesting to reproduce these experiments in human cardiomyocytes, for example the AC12 cell line.

3. Now, regarding the in vivo data in mice, the biocompatibility was only assessed in a limited number of animals (n= 4/group), which seems too little for conduct appropriate statistical test. In addition, data on the MgHCF NCs' ADME properties are lacking to properly evaluate potential adverse effects. How fast the agent is absorbed after i.p. injection. What are the plasma halftime and Cmax. To what tissues/organs is the agent distributed ? What is the proportion of the injected compound reaching the target organ (heart) and what is the mechanism of uptake by cardiac cells. For a better risk assessment of the exposure to MgHCF NCs, it is also mandatory to know the expected efficacy dose. This could help in the determination of a margin of safety. Finally, the rationale behind the selection of doses (both for MgHCF NCs and dexrazoxane) used in the in vivo experiments should clearly be stated.

4. In the safety assessment of MgHCF NCs in vivo (mice), I regret that the authors did not investigate the potential effect of the magnesium released from nanomedicine during iron uptake. Indeed, a magnesium overload is known to cause cardiovascular effects, including cardiac cytotoxicity and hypotension.

In most of the in vitro and in vivo assays (except maybe for the biocompatibility) presented in the study, the number of replicates and statistical tests are appropriate.

Analytical approach

As already mentioned, the analytical methods used in the physico-chemical characterization of the nanomedicine are outside the scope of my expertise. Regarding other analytical approaches, I only have one specific comment related to the assessment of cardioprotection using echocardiography. I do believe that those registered cardiac parameters are not sufficient for a comprehensive risk assessment. In particular, due to the release of Mg⁺⁺ ions in the cardiac cells during iron uptake by the nanocatalyst, I would recommend to add ECG-like recordings.

Suggested improvements

As already mentioned, additional information should be provided regarding :

1. Rationale for the selection of the doses (MgHCF NCs and dexrazoxane) used for the in vivo experiments
2. Better characterization of MgHCH NCs' ADME properties
3. Exposure of mature human cardiomyocytes to MgHCF Ncs (AC12 cell line)
4. Determine potential release of Mg²⁺ from nanocatalyst in blood compartment or in other tissues

5. Discuss the potential risk of Mg⁺⁺ release in cardiomyocytes during iron uploading in the nanocarrier

Minor changes :

1. Lines 41, 48 428 : replace “systematic” by “systemic”
2. Line 75 : replace “suffer from. The” by “suffer from the “
3. Line 81 : the sentence “remains limited chelation strengthen and capability” is not clear, please rephrase
4. Line 112 : replace “administration for during the“ by “administration during the”
5. Line 195 : replace “and the simultaneous the magnesium “ by “and simultaneously the magnesium “
6. Line 352 : replace “we next compare “ by “we next compared”
7. Line 397 : replace “ameliorated MgHCF NCs “ by “ameliorated by MgHCF NCs”
8. Line 620 : reword “has been being”
9. Line 739: replace “cardiomyocytes” by “cardiomyoblasts”

Clarity and context

The text is precise and perfectly understandable. The legends of the figures are self-sufficient to understand the graphics without necessary return to the text.

References

The references are appropriate and satisfactorily justify the points to which they are linked in the text.

Reviewer's expertise

As already mentioned, I do not have the expertise concerning the physicochemical aspects of characterization of the nanomaterial developed and evaluated in this study.

Reviewer #1 (Remarks to the Author):

In this paper, MgHCF nanocatalysts (NCs) have been synthesized through a PVP-directed self-assembly method. The NCs possess potent ferrous capturing and antioxidation functions for cardioprotection during tumor chemotherapy. The viability of the cardiomyocytes impaired by DOX could be effectively protected by the MgHCF NCs. The MgHCF NCs also exhibit excellent *in vivo* cardioprotection effect via attenuating the cardiotoxicity and the harmful side-effect due to the DOX contamination without deteriorating the anti-tumor effect of the chemodrug. The MgHCF NCs is novel and effective for the cardioprotection application. Their *in vivo* and *in vitro* effects have been thoroughly demonstrated. The paper is well organized. But the characterizations of MgHCF NCs are insufficient. This paper can be published after the following revisions:

Response: Thank you for your positive recommendations. We appreciate the reviewer's time and efforts in reviewing the manuscript. We have supplemented a number of experiments and revised the manuscript according to the editors' and reviewers' suggestions and we hope the major revision have addressed your valuable comments. Please find your point-by-point responses below.

(1) In Scheme 1 and Figure S1, what do the green spheres and light blue thorny particles mean? They should be indicated in the figures.

Response: Thank you for your suggestion. The thorny molecules represent the Polyvinyl Pyrrolidone (PVP) polymer while the green spheres represent the formulated MgHCF NCs. For higher clarity, we have revised all the MgHCF-related figures in the revised Manuscript and Supporting Information.

(2) The detailed structure of MgHCF NCs is not clearly clarified. Actually, the schematic structure in Figure 1a is not supported by the data. More evidence should be supplied to clarify the structure of MgHCF NCs.

Response: Thank you for your comment. To better clarify the structure of MgHCF NCs,

we have supplemented additional characterizations (XRD patterns and FTIR spectra) of MgHCF NCs and other material counterparts during the revision. From the supplemented XRD pattern of PVP, it is clear that the MgHCF NCs displays a similar pattern as PVP polymer (**Figure S4**). From the supplemented FTIR spectra, we can find that the spectrum of MgHCF NCs contains all peaks belonging to PVP polymer with additional C≡N stretching vibrations (updated **Figure 1e**). Based on the interactions of PVP-Mg²⁺ as well as PVP-ferricyanide ions, we therefore confirm that the MgHCF NCs are amorphous nanoparticles assembled from the ferricyanide and magnesium ions through the interaction with polyvinyl pyrrolidone polymer. Additionally, we have also revised the graphic of MgHCF NCs for better clarification throughout the manuscript.

(3) In Figure 1d and 1e, the XRD patterns and FTIR spectra of pure PVP and MgHCF NCs should be supplied for better comparison.

Response: Thank you for your suggestions. During the revision, the XRD patterns and FTIR spectra of PVP, MgHCF NCs as well as the potassium ferricyanide powders have been obtained (**Figure S4 in the revised Supporting Information**, updated **Figure 1e** in the revised **Manuscript**).

From the supplemented XRD pattern of PVP, we have found that the MgHCF NCs displays a similar pattern as PVP polymer (**Figure S4**). From the supplemented FTIR spectra, it is clear that the spectrum of MgHCF NCs contains all peaks attributable to PVP polymer with additional C≡N stretching vibrations (updated **Figure 1e**). Based on the interactions of PVP-Mg²⁺ as well as PVP-ferricyanide ions, we therefore confirm that the MgHCF NCs are amorphous nanoparticles assembled from the ferricyanide and magnesium ions through the interaction with polyvinyl pyrrolidone polymer.

(4) The ferrous capturing ability of MgHCF NCs should be proved under the interference of intracellular biomacromolecules. Thus the Mg²⁺ and Fe²⁺ exchange experiments should be performed in the presence of cell lysis to mimic the in vivo environments.

Response: Thank you for your comment and suggestion. To better imitate the

intracellular environment containing abundant biomacromolecules, the authors have employed RPMI full media with 10% fetal bovine to initiate the cationic exchange experiment. From the ICP results of the magnesium and iron concentrations, we can find that the ferrous capturing ability of MgHCF NCs will not be affected by the biomacromolecules interference. Data are presented in **Figure S5** in the revised Supporting Information.

(5) After ferrous capturing, how do the antioxidation performances of MgHCF NCs change?

Response: Thank you for your suggestion. To address your concern, we initially synthesized MgHCF NCs followed by the ferrous addition to form PB NPs. We then evaluated the antioxidation performance of the formulated PB NPs by using a dissolved oxygen meter as well as a SOD-assay kit. Supplemented data has been presented in **Figure S6**. Relevant discussion has been supplemented in **Page 9** in the revised Manuscript.

(6) In Figure S14, why do DCFH-DA-stained cardiomyocytes treated with DOX supplemented with MgHCF at the concentration of 80 ppm show much stronger fluorescence than that treated at the concentration of 40 ppm? And in Figure S15, DCFH-DA-stained cardiomyocytes treated with DOX supplemented with DXZ at the concentration of 25 ppm still showed much stronger green fluorescence, it cannot say that the oxidative stress is significantly relieved.

Response: Thank you for your comment. From Figure S14 (**now Figure S18**), the fluorescence intensity of cells treated with 40 ppm MgHCF is slightly stronger than those treated with 80 ppm MgHCF. Nevertheless, the confocal microscopic images present qualitative data of the intracellular oxidative stress. Such differences are most probably associated with by the heterogeneity of the region of interests. As for Figure S15 (**now Figure S19**), we agree with the reviewer that supplementation with DXZ at the concentration of 50 ppm rather than 25 ppm could relieve the oxidative stress. Therefore, we have revised the statements as follows:

“To our surprise, the co-incubation of DOX-treated cardiomyocytes with MgHCF ([Fe] = 40 $\mu\text{g ml}^{-1}$) and DXZ (50 $\mu\text{g ml}^{-1}$) could significantly relieve the oxidative stress inside the cardiomyocytes (Figure 4d-e, Figure S18-S19).”

(7) Beside the FerroOrange staining method, could the authors supply more quantitative data to measure the intracellular ferrous concentration?

Response: Thank you for your comment and suggestion. During the revision, we have quantitatively assayed the intracellular iron concentration by ICP method for AC16 cardiomyocytes subjected to different treatments. Cells in control, DOX treated, D_DXZ, and D_MgHCF groups were harvested, washed, counted and homogenized for elemental detections. We then present the assayed iron concentrations based on the number of cells (per 10^6 cells) for these groups. We have found that DOX treatments specifically increase the intracellular iron concentrations from 0.099 ± 0.01 to 0.780 ± 0.07 $\mu\text{g}/\text{million cells}$. While therapeutic treatments of DXZ and MgHCF NCs could effectively lower the intracellular iron concentration to 0.261 ± 0.02 $\mu\text{g}/\text{million cells}$ and 0.008 ± 0.01 $\mu\text{g}/\text{million cells}$ respectively (**Figure S25** in the revised **Supporting Information**, discussion on **Page 16** in the revised **Manuscript**).

Reviewer #2 (Remarks to the Author):

Hou et al investigated the Magnesium Hexacyanoferrate Nanocatalyst attenuates chemodrug-Induced cardiotoxicity through ferromodulation-driven anti-apoptosis. Excellent in vitro and in vivo cardioprotection performances of MgHCF NCs have been demonstrated and the underlying intracellular ferrous traffic regulation mechanism has been explored in detail. The marked cardioprotective effect and biocompatibility render MgHCF NCs to be a highly promising and clinically transformable cardioprotective agent to be employed during cancer treatments. This is an interesting finding in the Dox induced cardiotoxicity treatment. However there are some questions need to be explained as follow:

Response: Thank you for your positive recommendations. We appreciate the reviewer's time and efforts in reviewing the manuscript. We have supplemented a number of experiments and revised the manuscript according to the editors' and reviewers' suggestions and we hope the major revision have addressed your valuable comments. Please find your point-by-point responses below.

Major:

1. The studies on the biological properties of MgHCF NCs is not sufficient, including the half-life of the drug in the blood, drug utilization, median lethal dose, and drug distribution in different organs after intraperitoneal injection of MgHCF NCs, especially the drug concentration in the heart and tumor should provided.

Response: Thank you for your comments and suggestions. we have supplemented a number of experiments regarding the pharmacokinetic properties of MgHCF NCs. In a median lethal dose investigation, we directly concentrated the MgHCF NCs for dose biosafety evaluation. Mice received intraperitoneal administration of MgHCF NCs (single dose) at an extremely high dose of 30 mg kg⁻¹ show an overall survival rate of 80 % while those administrated with lower doses of 15, 10 and 7.5 mg kg⁻¹ all survived for two-weeks with healthy status. It could be calculated that the median lethal dose of MgHCF NCs for mice is about 52.5 mg kg⁻¹ (**Figure S40, discussion on Page 24 in**

the revised Manuscript). Higher doses of MgHCF NCs are not applicable due to the solubility limit and stability issue.

We also agree with the reviewers that pharmacokinetic performance of MgHCF NCs is the most pivotal evaluation for their clinical translation. Therefore, we have supplemented the half-terminal time (**Figure S41a**) and tissue biodistribution experiments of MgHCF NCs (**Figure S41b**). We found that upon MgHCF NCs administration, MgHCF NCs are mainly distributed into liver, spleen and lung in 2 h post-injection. In 12 h and 24 h post-injection, the overall distribution amounts of Mg²⁺ into these major organs were gradually decreased. Specifically, the distributed percentages of Mg²⁺ into heart were determined to be 7.16 ± 0.39 ID %/g, 5.39 ± 1.77 ID %/g and 1.95 ± 0.27 ID %/g respectively. For tumor accumulation, 2.84 ± 0.84 ID %/g, 1.23 ± 0.45 ID %/g and 1.20 ± 0.52 ID %/g could be determined in 2 h, 12 h and 24 h post-injection, respectively. Relevant discussion has been supplemented in the revised Manuscript (**Page 25**).

We have plotted the time-course plasma concentration of Mg²⁺ post MgHCF administration to determine the plasma half-life of MgHCF NCs. Following the exponential curve fitting, we have calculated the half terminal life of MgHCF NCs to be 1.59 h. At t = 0, plasma concentration of Mg²⁺ was determined to be 26.4 µg ml⁻¹, the drug utilization of MgHCF NCs was then calculated to be 55.0 % using the following equation based on an averaged mouse weight of 20 g and total blood volume = 2 mL (**Page 25**):

$$w = C_0 / C_{ID} = C_0 / (ID * m / V) \times 100 \%$$

where w represents the drug utilization; C₀ represents the plasma concentration of Mg²⁺ at t = 0; ID represents the injection dose, i.e., 4.8 mg kg⁻¹; m represents the weight of mice; V represents the total circulation blood volume of a mice.

2. Long-term side effects of MgHCF NCs on the number of red blood cells and hemoglobin should be investigated?

Response: Thank you for your suggestion. We have supplemented the blood routine assays for mice in one-month post administration with multiple doses of MgHCF NCs.

The results have been supplemented as **Figure S31 in the revised Supporting Information**. From the blood routine assays, unaffected red blood cells and hemoglobin levels could be determined, revealing the satisfactory long-term biosafety of MgHCF NCs.

3. The important question in this study is, why the authors didn't study cardiomyocyte ferroptosis, but apoptosis? There have been many reports about DOX induced myocardial ferroptosis. If MgHCF NCs can inhibit iron ions, the main target should be ferroptosis.

Response: Thank you for your comment. According to the previous literatures, several programmed cell death pathways such as apoptosis and ferroptosis would participate in the pathology of doxorubicin-induced cardiotoxicity. We have investigated the cell apoptosis origin by evaluating intracellular oxidative stress, as well as the pro-apoptosis regulations in mRNA and protein aspects. In addition, intracellular abnormal iron accumulations and ferroptosis-associated biomarkers have been studied. As the intracellular iron regulations play an important role in both cell apoptosis and ferroptosis, we have performed detailed investigations on both apoptosis and ferroptosis.

From our cellular mRNA sequencing results, we can observe combined cell death pathways for DOX-contaminated cardiomyocytes such as apoptosis, ferroptosis, autophagy and mitophagy, with apoptosis being the most dominant induced by DOX chemodrug. Although the transcriptome regulations by MgHCF contribute to the enrichment of ferroptosis KEGG pathway with significance (**Figure 4**), the scored gene ratio (0.779 %) remained relatively low (compared to 2.336 % of apoptosis). According to the highly up-regulated GPX4 which encodes glutathione peroxidase for lipid peroxidation clearance in DOX pathology, the pathological regulation of intracellular iron is believed to be less significant to cause prevailing cell ferroptosis judged from the non-destruction of anti-lipid peroxidation system. Based on the above considerations, we believed that apoptosis is still the main cell death pathway during DOX-induced pathology.

4. The H9c2 cells used in this study are derived from myoblast cells, which are closer to those of skeletal muscle cells. H9c2 cells have the ability to proliferate, which is significantly different from cardiomyocytes. So it is recommended that the author repeat the relevant experiments with primary cardiomyocytes.

Response: Thank you for your suggestion and comment. We have repeated the relevant experiments using human cardiomyocyte AC16 cell line. The supplemented experiments include the DOX-challenged toxicity (**Figure S8a-b**); cell rescue profile by DXZ or MgHCF NCs (**Figure S8c-e**) and confocal microscopic images of intracellular ferrous identification (**Figure S24**) as well as the intracellular oxidative stress (**Figure S23**). Relevant discussions have been supplemented in the revised Manuscript (**Page 11, 16**). Based on the above supplemented experiments, the same conclusion could be drawn for the AC16 cardiomyocytes as the H9c2 cardiomyoblasts. These experiments further validate that MgHCF NCs could effectively rescue the DOX-challenged cardiotoxicity with high biocompatibility in both cardiomyoblasts and cardiomyocytes.

Minor:

1. The quality of Fig 4J Western blot is too poor.

Response: Thank you for your suggestion. Western blot for Slc40a1, Tfrc, Nrf2, FTH1 and their corresponding references have been reconducted. Higher quality of the bands could be observed in the updated **Figure 4J** in the revised Manuscript. Protein quantification has also been updated (**Figure S27**).

The resolution of Figure 4J has been improved and updated.

2. Fig 6J LVEF% should not use normalized data, but should use raw data.

Response: Thank you for your comment. Raw LVEF values have been replotted as Figure 6g. To better indicate the profile trend of LVEF, the authors kindly keep the normalized LVEF plot as Figure 6h.

3. The caspase-3 in Figure is incorrectly labeled, it should be cleaved caspase-3.

Response: Thank you for your reminder. The labeling has been corrected in the revised manuscript.

4. In the in vivo data, n=4, which does not meet the statistical requirements.

Response: Thank you for your comment. During the revision period, we have supplemented several biocompatible experiments (**Figure S30, S31**), investigation (**Figure S39**), survival experiment (**Figure S40**), pharmacokinetic experiment (**Figure S41**) with a higher replication number (n = 10) to meet the statistical requirements.

5. The animal survival curve is that n=6-5 does not meet the statistical requirements. Generally, the number of animals in each group should be 10-20.

Response: Thank you for your comment. The survival experiment has been reconducted with a replication number of 10. Results have been updated in **Figure 5c** in the revised manuscript.

6. The detection data on autophagy, mitophagy, apoptosis and ferroptosis of cells and animals are insufficient.

Response: Thank you for your comment and suggestion. The present manuscript has been focused on the ferrous traffics and oxidative-associated pathologies during the cardiotoxicity and cardioprotection by MgHCF NCs and DXZ. The authors have elaborated to characterize the possible markers and proteins that are relevant to ferroptosis and apoptosis, including the intracellular free ferrous detection, intracellular oxidative stress and key protein expressions *etc.* We may have detected the DOX-induced autophagy and mitophagy pathology during the cardiotoxicity investigation using mRNA-seq. However, the main thesis of the present work is on the ferromodulation enabled by MgHCF NCs. We appreciate your understanding.

In short, this study is interesting, but not sufficient to be published in NC. More

functional studies and pharmacological studies data should provided to support the key results.

Response: Thank you for your kind comment. Following your suggestions, we have supplemented major functional and pharmacological studies as presented above. We hope that the major revision could well-address your concern.

Reviewer #3 (Remarks to the Author)

In cancer patients, treatment with anthracyclines (i.e., doxorubicin) is very often jeopardized by drug-induced cardiotoxicities mostly caused by excessive cellular oxidative stress. Here, the authors clearly demonstrate that the accumulation of radical species is catalyzed by Fenton-like reactions consecutive to alterations in iron transport and report on the development of a new nanomedicine, called MgHCF NCs, specifically designed to capture iron and hence reduce the iron-associated risk. After demonstrating its efficacy in eliminating the cytotoxic radical species in vitro, the authors showed the biocompatibility and cardioprotection of MgHCF NCs in vitro (H9C2 cardiomyoblasts) and in vivo (mice). Finally, in a tumor mouse model, the authors demonstrate that MgHCF NCs do not impair the anticancer efficacy of doxorubicin while significantly reducing its cardiac side effects. Based on convincing data, they conclude that MgHCF NCs displays all properties for a promising cardioprotective agent during cancer treatments.

Response: Thank you for your comprehensive comments and positive recommendations. We appreciate the reviewer's time and efforts in reviewing the manuscript. We have been supplemented a number of experiments and revised the manuscript according to the editors' and reviewers' suggestions and we hope the major revision have addressed your valuable concerns. Please find your point-by-point responses below.

Validity

The general framework of the research is based on a proven clinical reality according to which patients exposed to anthracyclines, such as doxorubicin, develop cardiotoxic damage during treatment, compromising the chances of success. This adverse effect is caused by the generation of reactive oxygen species, creating oxidative stress induced by the anticancer agent. The phenomenon would be amplified by an alteration of iron transport in heart cells, creating an environment conducive to Fenton reactions catalyzing the genesis of free radicals. This hypothesis has been partially confirmed in

previous studies showing the partially protective effect of an iron chelator, dexrazoxane, the only cardioprotective agent used clinically to date in this type of chemotherapy. In this publication, the authors present convincing results obtained in various in vitro and in vivo models showing the superior efficacy to dexrazoxane of a new nanomaterial, MgHCF NCs. The data clearly show that this nanocatalyst induces excellent cardioprotection thanks to its iron capture (by replacement of initially trapped magnesium ions) and antioxidant properties. In vitro, according to their transcriptomic and proteomic data acquired in the H9C2 line of cardiomyoblasts, the authors show that this nanocatalyst significantly increases the chances of survival of cardiac cells exposed to doxorubicin by reducing both deregulation of iron trafficking and cell apoptosis. In vivo (mice), this cardioprotective effect was partially confirmed by cardiac echocardiography studies. Finally, the authors presented in vivo data in mice showing the verified biocompatibility of the new agent, during repeated exposures over several weeks and also that the latter does not interfere with the anticancer capacities of doxorubicin in a mouse model implanted with a subcutaneous tumor. The encouraging results obtained for the new nanomaterial in terms of cardioprotection are supported by the fact that in the majority of in vitro and in vivo tests evaluated in this study, the latter performed better than dexrazoxane, used here as a positive benchmark.

Overall, the authors' conclusions (claiming that their new MgHCF Ncs nanomaterial is one of the most promising cardioprotective agents for clinical use) are appropriately supported by the data, well justified and reliable.

Response: Thank you for your comprehensive comments on the validity of our work. We appreciate the reviewer's time and efforts in reviewing the manuscript.

Significance

To the best of my knowledge, the authors' working hypotheses on the cellular mechanism leading to doxorubicin-induced cardiotoxicity, as well as their conclusions regarding the efficacy of their new nanocatalyst as a promising cardioprotective agent during cancer therapy are correct. I am not aware of any publication supporting contradictory data.

Response: Thank you for your positive recognition of the significance of our work.

Data and methodology

First, I would like to mention that I do not have the expertise concerning the physicochemical aspects of characterization of the nanocatalyst at the center of this publication. I will therefore not comment on this part of the results. On the other hand, I feel comfortable with the evaluation of in vitro and in vivo tests of biocompatibility and efficacy of the product. Here are my main comments on this part :

1. Concerning the in vitro evaluation of MgHCF NCs antioxidant properties, I am not convinced with the use of “multi-enzymatic catalytic performance” (see line 196) as well as with the “SOD superoxide dismutase- and catalase-like catalytic activities” (see for example lines 30) of MgHCF NCs. Such wording should be avoided since the agent can indeed induce those effects but lacks enzymatic activity.

Response: Thank you for your comments and suggestions. In the revised manuscript, we have revised “SOD-like activities” to “superoxide radical dismutation activities”. We also rephrased “CAT/catalase like catalytic activities” to “H₂O₂-decomposition activities” following the reviewer’s suggestions.

2. In the evaluation of MgHCF NCS’ cardioprotective properties, the authors used both in vitro and in vivo assays. In the in vitro cellular experiments, they selected the H9C2 cell line and inappropriately called them “cardiomyocytes” (line 739). In fact, H9C2 cells are cardiomyoblasts which exhibit most of the phenotypic characteristics of mature heart cells except the contractile properties. As a result, they are less dependent on oxidative phosphorylation and mitochondrial activity, the targets of doxorubicin studied here. It would be interesting to reproduce these experiments in human cardiomyocytes, for example the AC12 cell line.

Response: Thank you for your suggestions and comments. We have repeated the relevant experiments using human cardiomyocyte AC16 cell line. The supplemented experiments include the DOX-challenged toxicity (**Figure S8a-b**); cell rescue profile

by DXZ or MgHCF NCs (**Figure S8c-e**) and confocal microscopic images of intracellular ferrous identification (**Figure S24**) as well as the intracellular oxidative stress (**Figure S23**). Discussions have been supplemented in the revised Manuscript (**Page 11, 16**). Based on the above supplemented experiments, the same conclusion could be drawn for the AC16 cardiomyocytes as the H9c2 cardiomyoblasts. These experiments further validate that MgHCF NCs could effectively rescue the DOX-challenged cardiotoxicity with high biocompatibility in both cardiomyoblasts and cardiomyocytes.

3. Now, regarding the in vivo data in mice, the biocompatibility was only assessed in a limited number of animals (n= 4/group), which seems too little for conduct appropriate statistical test. In addition, data on the MgHCF NCs' ADME properties are lacking to properly evaluate potential adverse effects. How fast the agent is absorbed after i.p. injection. What are the plasma halftime and Cmax. To what tissues/organs is the agent distributed ? What is the proportion of the injected compound reaching the target organ (heart) and what is the mechanism of uptake by cardiac cells. For a better risk assessment of the exposure to MgHCF NCs, it is also mandatory to know the expected efficacy dose. This could help in the determination of a margin of safety. Finally, the rationale behind the selection of doses (both for MgHCF NCs and dexrazoxane) used in the in vivo experiments should clearly be stated.

Response: Thank you for your suggestions. We have supplemented a new set of in vivo biocompatibility evaluation (n = 10), in which a body weight profile of mice was recorded. At the end of the evaluation, major blood biochemical indexes and histology examinations were investigated. These supplemented data are presented as **Figure S30** in the revised Supporting Information. Relevant discussion has been supplemented in **Page 19** in the revised Manuscript.

To evaluate the ADME properties of MgHCF NCs, we have supplemented detailed experiments regarding the pharmacokinetic properties of MgHCF NCs (half-terminal time experiment, tissue biodistribution experiment, etc.) (**Figure S41**). We found that upon MgHCF NCs administration, MgHCF NCs were mainly distributed

into liver, spleen and lung in 2 h post-injection. In 12 h and 24 h post-injection, the overall distribution amounts of Mg^{2+} into these major organs are gradually decreased. Specifically, the distributed percentages of Mg^{2+} into heart were determined to be 7.16 ± 0.39 ID %/g, 5.39 ± 1.77 ID %/g and 1.95 ± 0.27 ID %/g respectively. For tumor accumulation, 2.84 ± 0.84 ID %/g, 1.23 ± 0.45 ID %/g and 1.20 ± 0.52 ID %/g could be determined in 2 h, 12 h and 24 h post-injection, respectively. Relevant discussion has been supplemented in the revised Manuscript (**Page 25**).

For plasma half-life of MgHCF NCs, we have plotted and fitted the time-course plasma concentration of Mg^{2+} post MgHCF administration. Following the exponential curve fitting, we have calculated the half terminal life of MgHCF NCs to be 1.59 h. At $t = 0$, plasma concentration of Mg^{2+} was determined to be $26.380 \mu\text{g ml}^{-1}$. For an averaged mice with m of 20 g, total blood volume = 2 mL, the drug utilization of MgHCF NCs was then calculated to be 54.96 % following the equation (**Page 25**).

$$w = C_0 / C_{ID} = C_0 / (ID * m / V) \times 100 \%$$

where w represents the drug utilization; C_0 represents the plasma concentration of Mg^{2+} at $t = 0$; ID represents the injection dose, i.e., 4.8 mg kg^{-1} ; m represents the weight of mice; V represents the total circulation blood volume of a mice.

The uptake mechanism of the nanomaterials by cardiomyocytes can be determined by the surface charge and chemical composition of the nanomaterials (Small, **2010**, 6(1): 12-21). Negatively charged MgHCF NCs may non-specifically bind to the cationic sites on the plasma membrane of the cardiomyocytes with subsequent endocytosis. It has also been indicated that spontaneous contraction of the cardiomyocytes could improve the internalization of the negatively charged nanomaterials due to the K^+ efflux and subsequent increased membrane potential (Physiological reviews, **2005**, 85(4): 1205-1253).

In the median lethal dose investigation, we used concentrated MgHCF NCs for dose biosafety evaluation. Mice received intraperitoneal administration of MgHCF NCs (single dose) at an extremely high dose of 30 mg kg^{-1} show an overall survival rate of 80 %, while those administrated with lower doses of 15, 10 and 7.5 mg kg^{-1} all survived for two-weeks. It could be calculated that the median lethal dose of MgHCF NCs for

mice is about 52.5 mg kg⁻¹ (**Figure S40, discussion on Page 24 in the revised Manuscript**). Higher doses of MgHCF NCs are not applicable due to the solubility limit and stability issue.

To rationalize the dose selection of MgHCF NCs, we have determined the effective in vitro dose of MgHCF NCs to the cells as follows by using a MgHCF NCs solution at 40 µg ml⁻¹. Based on an averaged mice weight of 20 g, total blood volume = 2 mL, and assuming a drug utilization of 20 %, we have determined an injection dose of 20 mg kg⁻¹. To perform multiple dosing, we have reduced the dose to 5 mg kg⁻¹ and the injection solution was finally calibrated to a magnesium concentration of 4.8 mg kg⁻¹. The following paper (PNAS, 2019, 116(7): 2672-2680) has been referenced for the dose selection of DXZ.

4. In the safety assessment of MgHCF NCs in vivo (mice), I regret that the authors did not investigate the potential effect of the magnesium released from nanomedicine during iron uptake. Indeed, a magnesium overload is known to cause cardiovascular effects, including cardiac cytotoxicity and hypotension.

Response: Thank you for your comments and suggestions. After careful literature survey, we have found that it is of great difficulties to in vivo monitor the magnesium release. We also agree with the reviewer's opinion that magnesium overload is potentially harmful to the cardiac functionalities. During MgHCF cardioprotection, magnesium ions have been released from the nanomedicine during iron uptake. Such an ion exchange occurs concurrently and equimolarly. Under the injection dose of MgHCF NCs of 4.8 mg kg⁻¹ d⁻¹, the highest Mg²⁺ flux could be obtained when all of the magnesium ions were released into the cardiomyocytes (i.e., 4.8 mg kg⁻¹). Therefore, we employed multiple doses of MgHCF NCs (4.8 mg kg⁻¹) or MgCl₂ (10 mg kg⁻¹) injections to assess the biosafety through echocardiography. In the present investigation, both MgHCF NCs and Mg²⁺ exhibit good cardiac biocompatibility during 7 days evaluation timeframe. According to the echocardiographic and electrocardiogram inspections, direct intraperitoneal administration of free Mg²⁺ at a dose not higher than 10 mg kg⁻¹ should be safe, revealing that cardiac toxicity or other abnormalities is

negligible under current injection doses of MgHCF NCs (**Figure S39 in the revised Supporting Information, Page 23 in the revised Manuscript**).

In most of the in vitro and in vivo assays (except maybe for the biocompatibility) presented in the study, the number of replicates and statistical tests are appropriate.

Response: Thank you for your comment and suggestions. The biocompatibility experiment has been reconducted with a replication number of 10 (**Figure S30**) as presented and described above. We hope that the revision could satisfy your consideration.

Analytical approach

As already mentioned, the analytical methods used in the physico-chemical characterization of the nanomedicine are outside the scope of my expertise. Regarding other analytical approaches, I only have one specific comment related to the assessment of cardioprotection using echocardiography. I do believe that those registered cardiac parameters are not sufficient for a comprehensive risk assessment. In particular, due to the release of Mg⁺⁺ ions in the cardiac cells during iron uptake by the nanocatalyst, I would recommend to add ECG-like recordings.

Response: Thank you for your comment and suggestions. In the supplemented cardioprotection experiment, we have combined the echocardiography and electrocardiogram inspections to support the analyses (**Figure S39 in the revised Supporting Information, Page 23 in the revised Manuscript**).

Suggested improvements

As already mentioned, additional information should be provided regarding :

1. Rationale for the selection of the doses (MgHCF NCs and dexrazoxane) used for the in vivo experiments

Response: Thank you for your comments. To rationalize the dose selection of MgHCF NCs, we have determined the effective in vitro dose of MgHCF NCs to the

cells as follows by using a MgHCF NCs solution at $40 \mu\text{g ml}^{-1}$. Based on an averaged mice weight of 20 g, total blood volume = 2 mL, and assuming a drug utilization of 20 %, we have determined an injection dose of 20 mg kg^{-1} . To perform multiple dosing, we have reduced the dose to 5 mg kg^{-1} and the injection solution was finally calibrated to a magnesium concentration of 4.8 mg kg^{-1} . The following paper (PNAS, 2019, 116(7): 2672-2680) has been referenced for the dose selection of DXZ.

2. Better characterization of MgHCF NCs' ADME properties

Response: Thank you for your suggestions. Comprehensive pharmacokinetic experiments have been supplemented to support the ADME properties of MgHCF NCs (**Figure S41**). Detailed descriptions have been supplemented in **Page 25** in the revised **Manuscript**.

3. Exposure of mature human cardiomyocytes to MgHCF NCs (AC12 cell line)

Response: Thank you for your suggestion and comment. We have repeated the relevant experiments using human cardiomyocyte AC16 cell line. The supplemented experiments include the DOX-challenged toxicity (**Figure S8a-b**); cell rescue profile by DXZ or MgHCF NCs (**Figure S8c-e**) and confocal microscopic images of intracellular ferrous identification (**Figure S24**) as well as the intracellular oxidative stress (**Figure S23**). Relevant discussions have been supplemented in the revised Manuscript (**Page 11, 16**). With the above supplemented experiments, the same conclusion could be drawn for the AC16 cardiomyocytes as the H9c2 cardiomyoblasts. These experiments further validate that MgHCF NCs could effectively rescue the DOX-challenged cardiotoxicity with high biocompatibility in both cardiomyoblasts and cardiomyocytes.

4. Determine potential release of Mg^{2+} from nanocatalyst in blood compartment or in other tissues

Response: Thank you for your comment and suggestions. In vivo plasma half-life experiment and biodistribution experiment have been supplemented during the revision.

These experiments could be employed to determine the potential pharmacokinetics of magnesium ions from nanocatalyst in blood compartment and other tissues. Detail data and descriptions have been supplemented in the revised Manuscript (**Page 25**) and Supporting Information (**Figure S41**).

5. Discuss the potential risk of Mg⁺⁺ release in cardiomyocytes during iron uploading in the nanocarrier

Response: Thank you for your comment and suggestions. We agree with the reviewer's opinion that magnesium overload is potentially harmful to the cardiac functionalities. During MgHCF cardioprotection, magnesium ions were released from the nanomedicine for iron uptake. Such an ion exchange occurs equimolarly. Under the injection dose of MgHCF NCs of 4.8 mg kg⁻¹ d⁻¹, the highest Mg²⁺ flux could be obtained when all of the magnesium ions were released into the cardiomyocytes (i.e., 4.8 mg kg⁻¹). Therefore, we employed multiple doses of MgHCF NCs (4.8 mg kg⁻¹) or MgCl₂ (10 mg kg⁻¹) injection to assess the biosafety through echocardiography. In the present investigation, both MgHCF NCs and Mg²⁺ exhibit good cardiac biocompatibility during 7 days evaluation timeframe. According to the echocardiographic and electrocardiogram inspections, direct intraperitoneal of free Mg²⁺ at doses not higher than 10 mg kg⁻¹ should be safe, revealing that cardiac toxicity or other abnormalities is negligible under current injection doses of MgHCF NCs (**Figure S39 in the revised Supporting Information, Page 23 in the revised Manuscript**).

Minor changes :

1. Lines 41, 48 428 : replace “systematic” by “systemic”

Response: Thank you for your comment. The word “systematic” has been revised to “systemic” throughout the manuscript during revision.

2. Line 75 : replace “suffer from. The” by “suffer from the “

Response: Thank you for your comment. The typo has been fixed in the revised

manuscript.

3. Line 81 : the sentence “remains limited chelation strengthen and capability” is not clear, please rephrase

Response: Thank you for your comment. The sentence has been rephrased to “Nevertheless, the chelation effect for ferrous ions is not strong enough compared to the chemical capturing into the hexacyanoferrate lattice.” in the revised manuscript.

4. Line 112 : replace “administration for during the“ by “administration during the”

Response: Thank you for your comment. This typo has been fixed in the revised manuscript.

5. Line 195 : replace “and the simultaneous the magnesium “ by “and simultaneously the magnesium “

Response: Thank you for your comment. This sentence has been fixed in the revised manuscript.

6. Line 352 : replace “we next compare “ by “we next compared”

Response: Thank you for your comment. This grammatical error has been fixed in the revised manuscript.

7. Line 397 : replace “ameliorated MgHCF NCs “ by “ameliorated by MgHCF NCs”

Response: Thank you for your comment. This sentence has been fixed in the revised manuscript.

8. Line 620 : reword “has been being”

Response: The phrase “has been being” has been revised to “has been” in the revised manuscript.

9. Line 739: replace “cardiomyocytes” by “cardiomyoblasts”

Response: Thank you for your suggestion. For the in vitro experiments using H9c2 cardiomyoblasts, the word “cardiomyocytes” has been replaced by “cardiomyoblasts” throughout the manuscript.

Clarity and context

The text is precise and perfectly understandable. The legends of the figures are self-sufficient to understand the graphics without necessary return to the text.

Response: Thank you for your positive comments.

References

The references are appropriate and satisfactorily justify the points to which they are linked in the text.

Response: Thank you for your positive comments.

Reviewer’s expertise

As already mentioned, I do not have the expertise concerning the physicochemical aspects of characterization of the nanomaterial developed and evaluated in this study.

Response: Thank you for your comprehensive comments and positive recommendations. We appreciate the reviewer’s time and efforts in reviewing the manuscript.

REVIEWERS' COMMENTS

Reviewer #1 (Remarks to the Author):

The authors have conducted additional experiments and significantly improved the paper. They also answered all my questions, thus this paper can be published as it is.

Reviewer #2 (Remarks to the Author):

The authors has responded to the all the questions comprehensively. This article can be published.

Reviewer #3 (Remarks to the Author):

Dear Authors,

I have read carefully the answers you gave to the remarks I made during the first evaluation of your manuscript. I was able to see that you had taken the time to redo the experiments suggested to validate certain critical points of the first version. I particularly appreciated the new results on the AC16 cardiomyocyte line and on the biocompatibility of your compound in terms of magnesium release. I am now more comfortable in accepting the publication of your study.

Response to Reviewers

Reviewer #1 (Remarks to the Author):

The authors have conducted additional experiments and significantly improved the paper. They also answered all my questions, thus this paper can be published as it is.

Response: We thank the reviewer for reviewing and recommending to our work.

Reviewer #2 (Remarks to the Author):

The authors have responded to the all the questions comprehensively. This article can be published.

Response: We thank the reviewer for reviewing and recommending to our work.

Reviewer #3 (Remarks to the Author):

Dear Authors, I have read carefully the answers you gave to the remarks I made during the first evaluation of your manuscript. I was able to see that you had taken the time to redo the experiments suggested to validate certain critical points of the first version. I particularly appreciated the new results on the AC16 cardiomyocyte line and on the biocompatibility of your compound in terms of magnesium release. I am now more comfortable in accepting the publication of your study.

Response: We thank the reviewer for reviewing and recommending to our work.